# Contribution Weights:
# A Geometrical Analysis of Self-Attention Transformers

**Harry Jake Cunningham** [* 1 2]  **Nicola Muça Cirone** [* 2]

## Abstract

Analyzing attention weights has become a standard approach for interpreting the information flow of Large Language Models (LLMs). However, this approach has significant limitations as it neglects the geometric properties of the value vectors being aggregated. To address this gap, we introduce *Contribution Weights*, a projection-based metric that quantifies a token's influence by accounting for it's attention weight, value magnitude, and directional alignment with the layer output. We demonstrate that contribution weights provide a more faithful measure of token importance, consistently outperforming attention-based metrics in identifying semantically critical tokens across different decoder-only models, tasks, and datasets. Further, our metric enables novel mechanistic analysis of *attention sinks*. While previous work characterized sinks as passive repositories for excess attention, we reveal they serve an active functional role, suppressing information through a convex relationship between sink rate and output norm, stabilizing representations by opposing the semantic drift of low-confidence tokens.

## 1. Introduction

Self-attention governs information mixing in Large Language Models (LLMs) (Devlin et al.; Brown et al.) by computing an input-dependent weighted sum of tokens via attention weights $\alpha_{ij}$ (Vaswani et al.). A common interpretability approach analyzes these weights directly, often implicitly treating large attention mass as evidence of functional importance. However, many prior works have questioned this *attention-as-explanation* view (Jain & Wallace;

[1]Department of Computer Science, University College London, London, UK. Work completed whilst at UCL. [2]Cartesia.AI, San Francisco, US. Correspondence to: Jake Cunningham <harry.cunningham@cartesia.ai>.

*Proceedings of the 43rd International Conference on Machine Learning*, Seoul, South Korea. PMLR 306, 2026. Copyright 2026 by the author(s).

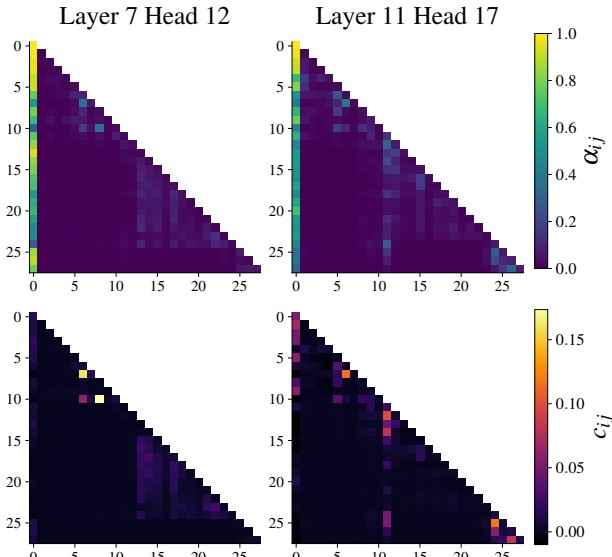

*Figure 1.* **Contribution weights**. **(Top)** Attention matrices and **(Bottom)** corresponding contribution weight matrices, at two randomly selected heads and layers, computed using LLAMA-3.1-8B. Whilst attention weights place a large amount of attention mass on the first token, examining the contribution weights reveals that these tokens contribute very little to the output.

Bastings & Filippova; Lopardo et al.), arguing that attention weights alone provide an incomplete account of information aggregation, as they do not account for the latent geometry of the vectors being aggregated (Kobayashi et al., a; Ferrando et al., a;b).

For interpretability analysis, multi-head attention (MHA) can be equivalently computed as a weighted sum of head-wise independent value vectors, aggregating information over the sequence dimension, where each value vector is projected by a learned linear map that shapes value geometry (Kobayashi et al., a; Elhage et al.). From this perspective, the influence of a token on the layer output depends not only on it's attention weight, but also on the magnitude of it's value vector and its alignment with the all other attended to tokens. Neglecting these geometric factors can lead to misleading conclusions by ignoring interference among vectors and by overestimating the importance of high-attention tokens. A prominent example is the study of *attention sinks*, where early tokens receive substantial attention despite con-

tributing little to the resulting representation once value norms and directional alignment are accounted for.

In this work, we introduce *contribution weights*, a projection-based metric that accounts for the full geometry of the attention vector sum and that faithfully captures token importance. We show that contribution weights decompose into the product of attention weight, relative magnitude, and directional alignment between value vectors and the output. Using this decomposition, we study the functional role of value geometry in self-attention, with particular focus on sink tokens.

Our key findings and contributions are:

- *Contribution weights faithfully measure token importance.* We demonstrate through causal interventions that contribution weights consistently outperform attention-based metrics in identifying critical tokens across multiple models, tasks, and datasets.

- *Directional alignment is critical to token contribution.* Incorporating the directional alignment of value vectors substantially improves the accuracy of contribution estimates across all sequence positions, revealing that geometric orientation plays a more important role than magnitude in determining token influence.

- *Value geometry drives the attention sink phenomenon.* Initial token value vectors are anti-aligned with other tokens in the sequence, and their contribution to the output is independent of their attention scores, explaining why high attention on these positions does not translate to semantic influence.

- *Sink tokens actively suppress information.* Contrary to existing characterizations of sinks as passive attention absorbers, we show that sink tokens function as active information suppressors that oppose the semantic direction of non-sink tokens, thereby nullifying semantic drift introduced by weak attention accumulation.

## 2. Related Work

**Geometry-Aware Attention Analysis.** To address the limitations of weight-based attention analysis, several value-aware measures incorporate geometric properties such as value norms (Kobayashi et al., a;b) and value-output distances (Ferrando et al., a). These methods more faithfully identify important tokens (Kobayashi et al., a;b) and enable effective token pruning (Ferrando et al., a) and KV cache compression (Devoto et al.; Guo et al., b; Shin et al.). Nevertheless, existing approaches capture only partial value geometry, neglecting effects such as inter-token interference, which we show to be crucial for understanding attention's functional role.

**Attention Sinks and Value Geometry.** Understanding how value geometry operates in tandem with attention weights is crucial for explaining phenomena like attention sink, where tokens receive disproportionately high attention despite carrying minimal semantic content (Xiao et al.). Recent work has characterized specific geometric properties of sink tokens, including high-norm residual stream hidden states (Sun et al.), small value norms (Kobayashi et al., a; Devoto et al.; Guo et al., b), and negative cosine similarity with non-sink hidden states (Shin et al.; Cancedda). These properties suggest sinks function as implicit bias terms (Sun et al.; Darcet et al.), absorbing excess attention and deactivating heads by driving output norms toward zero (Bondarenko et al.; Guo et al., a), slowing the rate of sequence mixing (Barbero et al.). However, despite these insights, the precise geometric mechanisms by which sinks affect sequence mixing, particularly through inter-token interference, remain under explored.

## 3. Background

In this section, we provide background on the attention mechanism, explicitly stating it's form as a weighted sum of projected value vectors.

### 3.1. Multi-Head Attention as Weighted Sum

Given an input sequence $\mathbf{X} \in \mathbb{R}^{L \times d}$ of length $L$ and dimension $d$, causal self-attention (Vaswani et al.) computes the output $\mathbf{Y} \in \mathbb{R}^{L \times d'}$ as a weighted summation over previous tokens $j \leq i$. At token position $i \in L$,

$$\boldsymbol{y}_i = \sum_{j=1}^{i} \frac{\exp(\boldsymbol{q}_i \boldsymbol{k}_j^T / \sqrt{d_h})}{\sum_{n=1}^{i} \exp(\boldsymbol{q}_i \boldsymbol{k}_n^T / \sqrt{d_h})} \boldsymbol{v}_j = \sum_{j=1}^{i} \alpha_{ij} \boldsymbol{v}_j \quad (1)$$

where $\alpha_{ij} \in [0, 1]$ are the scalar *attention weights* summing to 1 along $j \leq i$, $\mathbf{Q}, \mathbf{K}, \mathbf{V} = \mathbf{X}\mathbf{W}_Q, \mathbf{X}\mathbf{W}_K, \mathbf{X}\mathbf{W}_V \in \mathbb{R}^{L \times d'}$ are the query, key and value projections, with $\boldsymbol{W}_Q, \boldsymbol{W}_K, \boldsymbol{W}_V \in \mathbb{R}^{d \times d'}$ learnable parameter matrices.

The output of MHA $\boldsymbol{o}_i \in \mathbb{R}^d$ is conventionally expressed as the concatenation of $H$ *independent* attention heads $\boldsymbol{y}_i^{(h)} \in \mathbb{R}^{d'}$, with $d = H \cdot d'$, followed by a single output projection $\boldsymbol{W}_O \in \mathbb{R}^{d \times d}$,

$$\boldsymbol{o}_i = \text{Concat}(\{\boldsymbol{y}_i^{(h)}\}_{h=1}^{H})\boldsymbol{W}_O \quad (2)$$

Partitioning the output projection matrix into $H$ blocks, $\boldsymbol{W}_O = \text{Concat}(\{\boldsymbol{W}_O^{(h)}\}_{h=1}^{H}) \in \mathbb{R}^{Hd' \times d}$, where $\boldsymbol{W}_O^{(h)} \in \mathbb{R}^{d' \times d}$, we can express the MHA output as a summation over attention heads (Kobayashi et al., a; Elhage et al.),

$$\boldsymbol{o}_i = \left[\boldsymbol{y}_i^{(1)}, \boldsymbol{y}_i^{(2)}, \ldots\right] \begin{bmatrix} \boldsymbol{W}_O^{(1)} \\ \boldsymbol{W}_O^{(2)} \\ \vdots \end{bmatrix} = \sum_{h=1}^{H} \boldsymbol{y}_i^{(h)} \boldsymbol{W}_O^{(h)} \quad (3)$$

From this perspective, MHA can be equivalently viewed as the sum of independently projected head outputs $\boldsymbol{o}_i^{(h)} := \boldsymbol{y}_i^{(h)} \boldsymbol{W}_O^{(h)} \in \mathbb{R}^d$, each additively contributing to the final representation.

### 3.2. Attention as Independent Linear Maps

Substituting the definition of attention (Equation (1)) into the additive representation of MHA (Equation (3)), one equivalently obtains $\boldsymbol{o}_i$ as

$$\boldsymbol{o}_i = \sum_{h=1}^{H} \sum_{j=1}^{i} \alpha_{ij}^{(h)} \boldsymbol{x}_j \boldsymbol{W}_V^{(h)} \boldsymbol{W}_O^{(h)} \quad (4)$$

This expression makes explicit that MHA factors into two independent head-level linear operations: (i) a weighted sum over the *sequence dimension* with scalar coefficients $\alpha_{ij}^{(h)}$, which adaptively determines how much token $j$ contributes to position $i$, and (ii) a *low-rank* linear map over the *feature dimension* $\boldsymbol{W}_V^{(h)} \boldsymbol{W}_O^{(h)} \in \mathbb{R}^{d \times d}$, which shapes the geometry of the $\boldsymbol{W}_O^{(h)}$-projected value vectors.

For ease of notation we refer to the inputs transformed by the product $\boldsymbol{W}_V^{(h)} \boldsymbol{W}_O^{(h)}$ as *projected value vectors*,

$$\tilde{\boldsymbol{v}}_j^{(h)} := \boldsymbol{x}_j \boldsymbol{W}_V^{(h)} \boldsymbol{W}_O^{(h)} = \boldsymbol{v}_j^{(h)} \boldsymbol{W}_O^{(h)} \in \mathbb{R}^d. \quad (5)$$

Combining these observations, the output of an attention layer $l$ in a transformer can be expressed as a weighted sum over projected value vectors:

$$\boldsymbol{o}_i^{(l)} = \sum_{h=1}^{H} \sum_{j=1}^{i} \alpha_{ij}^{(l,h)} \tilde{\boldsymbol{v}}_j^{(l,h)} \quad (6)$$

where $\tilde{\boldsymbol{v}}_j^{(l,h)} = \hat{\boldsymbol{h}}_j^{(l-1)} \boldsymbol{W}_V^{(l,h)} \boldsymbol{W}_O^{(l,h)}$ are the projected value vectors, and $\hat{\boldsymbol{h}}_j^{(l)} = \text{Norm}^{(l)}(\boldsymbol{h}_j^{(l)})$ are the normalised hidden state from the preceding layer.

This formulation of MHA highlights two key properties: (i) attention constructs hierarchical representations through successive stages of aggregation— value vectors are weighted and summed to form head outputs, which are in turn summed to produce layer outputs, (ii) attention explicitly shapes the geometry of value vectors, an often overlooked property that is central to our analysis of token contributions in the following section.

## 4. Contribution of Attention

We now introduce *contribution weights*, a projection based metric to analyse the behaviour of the attention mechanism, that accounts for both the magnitude of the attention weights and the geometry of value vectors.

### 4.1. Determinants of Vector Summation

Consider a weighted summation of $N$ vectors $\boldsymbol{v}_i \in \mathbb{R}^d$, each scaled by a scalar $\alpha_i \in \mathbb{R}$:

$$\boldsymbol{y} = \sum_{i=1}^{N} \alpha_i \boldsymbol{v}_i. \quad (7)$$

The resulting $\boldsymbol{y} \in \mathbb{R}^d$ is determined by three main factors:

1. **Magnitude of $\alpha_i$.** Each coefficient $\alpha_i$ scales its corresponding vector $\boldsymbol{v}_i$, modulating its relative influence. Under SoftMax attention, $\alpha_i \in [0, 1]$: $\alpha_i = 0$ implies no contribution, $0 < \alpha_i < 1$ attenuates the effect, and $\alpha_i = 1$ preserves the full vector.

2. **Norm of $\boldsymbol{v}_i$.** Even when $\alpha_i$ values are equal, vectors with larger norms contribute more strongly, since $\|\alpha_i \boldsymbol{v}_i\| = |\alpha_i| \|\boldsymbol{v}_i\|$. Hence, the combined influence of $\alpha_i$ and $\|\boldsymbol{v}_i\|$ determines each vector's effective strength in the summation.

3. **Alignment of $\boldsymbol{v}_i$.** The relative orientation of vectors determines the degree of interference in the summation. When pairwise cosine similarities $\cos(\boldsymbol{v}_i, \boldsymbol{v}_j) > 0$, the vectors reinforce one another, producing constructive interference and a larger resultant norm. When $\cos(\boldsymbol{v}_i, \boldsymbol{v}_j) < 0$, opposing directions lead to destructive interference, attenuating or cancelling the combined output.

### 4.2. Contribution weights

To better understand the aggregation behaviour of MHA, we introduce *contribution weights*, a projection-based measure that accounts for a token's attention weight, it's norm and directional alignment with the output. We show that our metric is naturally suited to evaluating the influence of representations across different levels of aggregation.

**Intra-Head Contribution Weights.** Consider the output at position $i$ from a *single* attention head,

$$\boldsymbol{o}_i = \sum_{j=1}^{i} \alpha_{ij} \tilde{\boldsymbol{v}}_j, \quad (8)$$

where $\alpha_{ij}$ are the attention weights and $\tilde{\boldsymbol{v}}_j$ the projected value vectors. The contribution of token $j$ depends on how strongly the weighted value vector $\alpha_{ij} \tilde{\boldsymbol{v}}_j$ projects onto the final output direction $\boldsymbol{o}_i$. We define the *intra-head contribution weights* as the normalized inner product:

$$\check{c}_{ij} := \frac{\langle \boldsymbol{o}_i, \alpha_{ij} \tilde{\boldsymbol{v}}_j \rangle}{\|\boldsymbol{o}_i\|^2} \in \mathbb{R}. \quad (9)$$

Contribution weights represent the signed fraction of the output's norm $\|\boldsymbol{o}_i\|$ attributable to each component in the

output's direction as they naturally sum to one across the sequence dimension,

$$\sum_{j=1}^{i} \check{c}_{ij} = \frac{\langle \boldsymbol{o}_i, \sum_{j=1}^{i} \alpha_{ij} \tilde{\boldsymbol{v}}_j \rangle}{\|\boldsymbol{o}_i\|^2} = \frac{\langle \boldsymbol{o}_i, \boldsymbol{o}_i \rangle}{\|\boldsymbol{o}_i\|^2} = 1. \quad (10)$$

When $\check{c}_{ij} > 0$, token $j$ reinforces the output direction, when $\check{c}_{ij} < 0$ it opposes it, and when $\check{c}_{ij} \approx 0$ it contributes orthogonally. This contrasts with attention weights ($\alpha_{ij} \in [0, 1]$), which only indicate selection strength without regard for directional alignment.

**Inter-Head Contribution Weights.** In multi-head attention, the output at position $i$ is the sum of projected value vectors from all heads,

$$\boldsymbol{o}_i = \sum_{h=1}^{H} \sum_{j=1}^{i} \alpha_{ij}^{(h)} \tilde{\boldsymbol{v}}_j^{(h)}, \quad (11)$$

We extend contribution weights to the multi-head setting by projecting each head's weighted value vectors onto the combined multi-head output direction $\boldsymbol{o}_i$,

$$\hat{c}_{ij}^{(h)} = \frac{\langle \boldsymbol{o}_i, \alpha_{ij}^{(h)} \tilde{\boldsymbol{v}}_j^{(h)} \rangle}{\|\boldsymbol{o}_i\|^2}. \quad (12)$$

Using the full multi-head output $\boldsymbol{o}_i$, rather than the head-specific output $\boldsymbol{o}_i^{(h)}$, captures both intra-head token influence and the relative contribution of each head to the combined output. This contrasts with attention weights, which are independently normalised within each head.

As a result, *inter-head contribution weights* are normalised jointly over heads and positions:

$$\sum_{h=1}^{H} \sum_{j=1}^{i} \hat{c}_{ij}^{(h)} = 1. \quad (13)$$

### 4.3. Decomposition of Contribution Weights

Using the cosine identity, contribution weights can be decomposed into three interpretable geometric factors[1],

$$c_{ij} = \frac{\langle \boldsymbol{o}_i, \alpha_{ij} \tilde{\boldsymbol{v}}_j \rangle}{\|\boldsymbol{o}_i\|^2} = \alpha_{ij} \frac{\|\tilde{\boldsymbol{v}}_j\|}{\|\boldsymbol{o}_i\|} \cos(\boldsymbol{o}_i, \tilde{\boldsymbol{v}}_j), \quad (14)$$

where $\cos(\boldsymbol{o}_i, \tilde{\boldsymbol{v}}_j) = \frac{\langle \boldsymbol{o}_i, \tilde{\boldsymbol{v}}_j \rangle}{\|\boldsymbol{o}_i\|\|\tilde{\boldsymbol{v}}_j\|}$ is the cosine similarity between the projected value and the output.

This decomposition makes apparent that a token's contribution is determined jointly by three independent factors:

---

[1]We omit the *iter/intra* notation difference here as the insight holds for both measures.

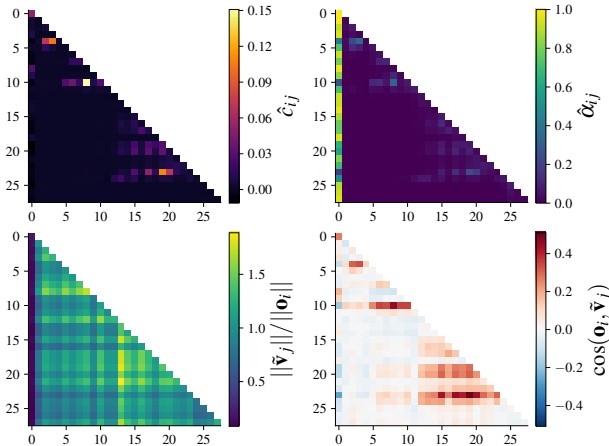

*Figure 2.* **Contribution weight decomposition.** We visualise the multiplicative components of the inter-head contribution weights $\hat{c}_{ij}$: (i) the attention weights $\alpha_{ij}$, (ii) the relative norm $\|\tilde{\boldsymbol{v}}_j^{(l,h)}\|/\|\boldsymbol{o}_i^{(l)}\|$, and (iii) the cosine similarity $\cos(\boldsymbol{o}_i^{(l)}, \tilde{\boldsymbol{v}}_j^{(l,h)})$. All quantities are computed on the same input for a randomly selected layer and head of LLaMA-3.1-8B.

1. **Attention weights** $\alpha_{ij}$: determines the selection strength; how much the token is attended to relative to other tokens in the sequence.

2. **Relative norm** $\frac{\|\tilde{\boldsymbol{v}}_j\|}{\|\boldsymbol{o}_i\|}$: scales the contribution by the magnitude of the projected value vector relative to the output norm.

3. **Cosine similarity** $\cos(\boldsymbol{o}_i, \tilde{\boldsymbol{v}}_j)$: identifies whether the token reinforces ($> 0$) or opposes ($< 0$) the output direction.

Together, these components provide a complete account of how tokens influence the attention output, capturing the full geometry of vector summation. As we'll explore later, this decomposition enables fine-grained analysis of how attention weights, value norms, and alignment interact both within and across attention heads to produce the final output.

## 5. Contribution Weights Reflect Token Importance

We first evaluate whether contribution weights produce more faithful explanations of a token's importance than existing attention-based metrics, paying particular attention to how attention weights, relative norms and cosine similarity contribute to increased faithfulness.

### 5.1. Token Importance

Given an input $\mathbf{X} \in \mathbb{R}^{L \times d}$, we compute the contribution of token $j$ to the output at position $i$ for layer $l$ and head $h$ according to a chosen measure $\gamma$ (e.g., attention weights $\alpha$ or contribution weights $c$). To quantify the importance

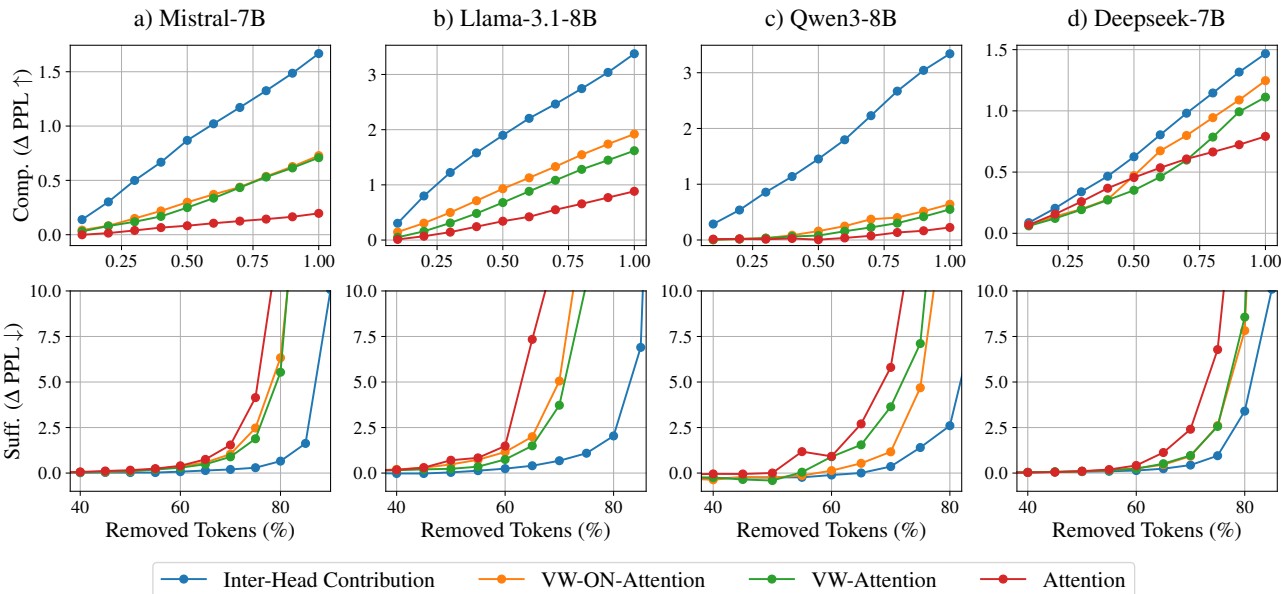

*Figure 3.* **Contribution weights most reliably predict token importance.** Change in perplexity (ΔPPL) on GSM8K under increasing token-removal rates, comparing four contribution measures: (1) contribution weights, (2) value-weighted output-normalised attention (VW-ON), (3) value-weighted attention, and (4) raw attention weights. **Top:** Removal of top percentiles. **Bottom:** Removal of bottom percentiles. Contribution weights outperform all other metrics for every removal rate, model and dataset, highlighting how they better reflect a token's importance.

of token $j$ according to $\gamma$, we define its *token-score* as its average contribution for all subsequent positions $i \geq j$:

$$\text{token-score}(\gamma)_j^{(l,h)} = \frac{1}{T - j + 1} \sum_{i=j}^{T} \gamma_{ij}^{(l,h)} \quad (15)$$

To evaluate a metric's faithfulness, we compute both its Comprehensiveness and Sufficiency (DeYoung et al., 2020).

**Comprehensiveness** (↑) measures the average change in model output after removing important tokens:

$$\text{Comp}(\gamma)_p = \frac{1}{N} \sum_{i=1}^{N} f(\tilde{\boldsymbol{v}}) - f(\tilde{\boldsymbol{v}}_{:1-p}^{\gamma}) \quad (16)$$

where $f$ stands for model perplexity, $\tilde{\boldsymbol{v}}$ are the original projected values at all layers, and $\tilde{\boldsymbol{v}}_{:q}^{\gamma}$ denotes the bottom-$q$ fraction projected values, at each layer, ranked by token-score$(\gamma)_j^{(l,h)}$. Larger drops indicate higher faithfulness.

**Sufficiency** (↓) measures the average change in model output after removing unimportant tokens:

$$\text{Suff}(\gamma)_p = \frac{1}{N} \sum_{i=1}^{N} f(\tilde{\boldsymbol{v}}) - f(\tilde{\boldsymbol{v}} \setminus \tilde{\boldsymbol{v}}_{:p}^{\gamma}) \quad (17)$$

Lower scores indicate higher faithfulness, as predictions change minimally with only important tokens retained.

We implement removal by masking projected values $\tilde{\boldsymbol{v}}_j^{(l,h)}$ to $\boldsymbol{0}$ at each layer $l$. For Comprehensiveness, we mask the top-$p$ fraction of tokens; for Sufficiency, we mask the bottom $p$. Masking is applied sequentially so that modifications at layer $l$ propagate through all subsequent layers.

### 5.2. Experimental Setup

**Measures.** We compare four measures of token importance $\gamma$, each incorporating progressively more information about value geometry: (i) Attention $\alpha_{ij}^{(l,h)}$, (ii) Value-weighted (VW) attention $\alpha_{ij}^{(l,h)}\|\tilde{\boldsymbol{v}}_j^{(l,h)}\|$, which scales attention by value magnitude, (iii) VW output-normalized (VW-ON) attention $\alpha_{ij}^{(l,h)}\|\tilde{\boldsymbol{v}}_j^{(l,h)}\|/\|\boldsymbol{o}_i^{(l)}\|$, which additionally normalizes by the output norm, and (iv) Absolute inter-head contribution (IHC) $|\hat{c}_{ij}|$, our proposed measure.

**Non-Semantic Tokens.** To focus on *semantic* influence, we exclude *sink tokens*, identified by near-perfect collinearity, a cosine similarity score $> 0.95$, with the [BOS] token, from removal. Deleting these semantically void tokens disproportionately degrades attention-based and value-weighted metrics (Xiao et al.), confounding our assessment of semantic importance. Section C presents results including sink token removal.

**Language Modeling.** We evaluate 4 models: (i) Mistral-7B (Jiang et al., 2023), (ii) LLaMA-3.1-8B (Touvron et al.), (iii) Qwen3-8B (Yang et al., 2025), and (iv) Deepseek-7B

(DeepSeek-AI et al., 2024), on GSM8K (Cobbe et al.) and FineWeb-Edu (Penedo et al.) with maximum sequence length 512. For each model we compute comprehensiveness and sufficiency as the change in perplexity ($\Delta$PPL) for different removal rates $k$.

**Downstream Tasks.** Using LLaMA-3.1-8B, we also assess the faithfulness of each measure on downstream tasks that require isolating relevant contextual information (Bick et al.; Merullo et al.): (i) MMLU (Hendrycks et al.), (ii) Winogrande (Sakaguchi et al.), and (iii) BoolQ (Clark et al.). We measure task accuracy before and after intervention and report sufficiency in Figure 4. A faithful metric should minimize accuracy degradation.

### 5.3. Results

**Contribution weights are more faithful than other attention-based metrics.** Across all models and removal rates, contribution weights consistently outperform attention-based metrics in both comprehensiveness and sufficiency (Figure 3), providing the clearest signal of token functional importance. On downstream tasks (Figure 4), contribution weights better preserve model performance under extreme sparsity, maintaining baseline accuracy at higher removal rates than attention-based alternatives. These results provide clear empirical evidence for the importance of value geometry in the attention mechanism.

**All geometric information contributes to faithfulness improvements.** Comparing importance metrics reveals a clear progression: *adding geometric information consistently improves faithfulness*. Across all tasks, raw attention weights perform worst, confirming that attention scores alone don't reliably indicate functional contribution. Performance improves when we multiply attention weights by value norms, and the marginal gain from VW to VW-ON in language modeling suggests that importance depends not only on intra-head geometry but also on each head's relative influence on the total layer output. Contribution weights, which incorporate angular information, prove most faithful. Adding relative alignment produces the largest performance increase among all metrics, suggesting that token influence is strongly affected by interference from other tokens.

## 6. Functional Role of Value Geometry

Having established that contribution weights are a faithful measure of a token's functional importance, we now use them to examine *what determines a token's contribution to the output*. We first decompose token contributions into the product of attention $\alpha_{ij}$, value magnitude $\|\tilde{\boldsymbol{v}}_j\|/\|\boldsymbol{o}_i\|$, and directional alignment $\cos(\boldsymbol{o}_i, \tilde{\boldsymbol{v}}_j)$, studying the their significance in shaping a token's contribution in Llama-

*Table 1.* **Regression analysis of contribution weight components.** We report the coefficient of determination ($R^2$) for linear regressions of contribution weights onto individual and combined multiplicative components. Each row corresponds to a different regression model: (1) the scalar attention weight $\alpha_{ij}$, (2) the relative norm of the projected value vector $\text{norm}_{ij} = \|\tilde{\boldsymbol{v}}_j\|_2/\|\boldsymbol{o}_i\|_2$, (3) the cosine similarity $\cos(\tilde{\boldsymbol{v}}_j, \boldsymbol{o}_i)$, and (4–6) products of these components. We evaluate regressions under three token groupings: (i) $j \in \{1, T\}$ (all tokens), (ii) $j = 1$ (initial tokens), and (iii) $j > 1$ (non-initial tokens). Higher $R^2$ values indicate that the corresponding component or combination explains more of the contribution weight variance.

| Components | $j \in \{1, T\}$ | $j = 1$ | $j > 1$ |
|---|---|---|---|
| $\alpha_{ij}$ | 0.055 | 0.079 | 0.449 |
| $\frac{\|\tilde{\boldsymbol{v}}_j\|}{\|\boldsymbol{o}_i\|}$ | 0.000 | 0.027 | 0.000 |
| $\cos(\boldsymbol{o}_i, \tilde{\boldsymbol{v}}_j)$ | 0.036 | 0.518 | 0.028 |
| $\alpha_{ij},\ \frac{\|\tilde{\boldsymbol{v}}_j\|}{\|\boldsymbol{o}_i\|}$ | **0.489** | 0.157 | 0.579 |
| $\alpha_{ij},\ \cos(\boldsymbol{o}_i, \tilde{\boldsymbol{v}}_j)$ | 0.305 | 0.611 | **0.834** |
| $\frac{\|\tilde{\boldsymbol{v}}_j\|}{\|\boldsymbol{o}_i\|},\ \cos(\boldsymbol{o}_i, \tilde{\boldsymbol{v}}_j)$ | 0.020 | **0.725** | 0.021 |

3.1-8B. Through causal interventions, we then examine the functional role of each component, revealing how the geometry of the sink token actively suppresses semantic tokens while concentrating contribution among them.

### 6.1. Component Importance

In Figure 2 we plot the contribution weights and each multiplicative component for a sample head. Noting the visual differences in distribution between components, we first examine how much of the variance in contribution can be explained by each subset of components and their interactions. Leveraging knowledge of its true functional form (Equation (14)), we analyse the direct multiplicative relationships between components. Specifically, for each subset of components,

$$S \subseteq \{\alpha_{ij}^{(l,h)},\ \|\tilde{\boldsymbol{v}}_j^{(l,h)}\|/\|\boldsymbol{o}_i^{(l)}\|,\ \cos(\boldsymbol{o}_i^{(l)}, \tilde{\boldsymbol{v}}_j^{(l,h)})\}, \quad (18)$$

we construct the multiplicative feature $\eta_S = \prod_{\eta \in S} \eta$ and then fit a simple linear model $c_{ij}^{(l,h)} = \beta_0 \eta_{S,ij}^{(l,h)}$ to the contribution weights. We then compute the coefficient of determination ($R^2$), which measures the variance in $c_{ij}^{(l,h)}$ explained by the multiplicative interaction of components in $S$. See Appendix D.2 for experimental details. In Table 1 we report results on FineWeb-Edu, observing a striking difference in results for initial or sink tokens ($j = 1$) versus tail tokens ($j > 1$).

**Sink tokens: Value geometry dominates.** For initial tokens, contribution is primarily determined by value geometry. The product of relative norm and cosine similarity serves as the strongest predictor ($R^2 = 0.725$), while attention weights alone have negligible explanatory power

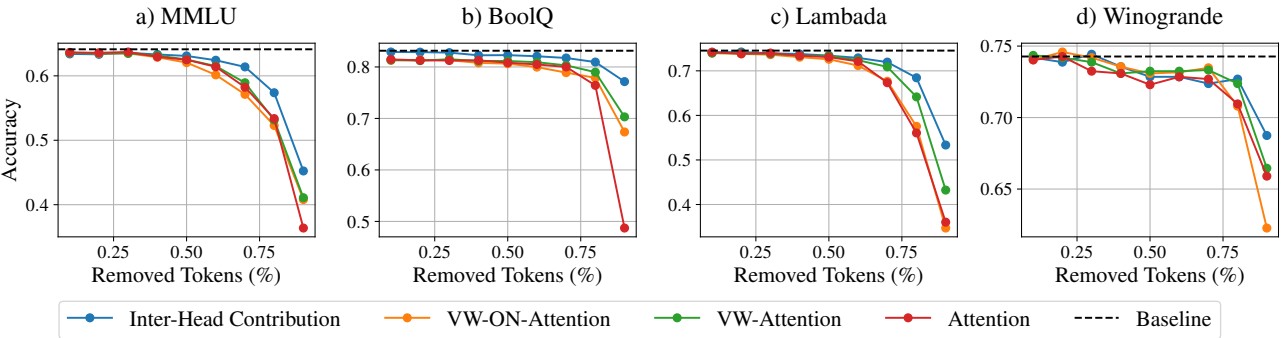

*Figure 4.* **Contribution weights maintain downstream performance under high token-removal rates.** Change in accuracy on MMLU, BoolQ, Lambada and Winogrande under increasing token-removal rates, comparing four contribution measures: (1) contribution weights, (2) value-weighted output-normalised attention (VW-ON), (3) value-weighted attention, and (4) raw attention weights. Contribution weights maintain baseline performance at a higher removal rate than other metrics.

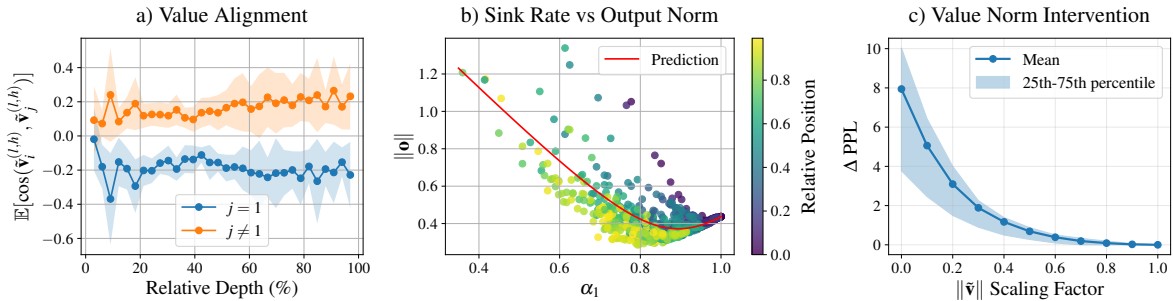

*Figure 5.* **Value vector geometry. (a)** Average alignment (cosine similarity) between the initial ([BOS]/sink) token's value vector and the tail tokens' value vectors as a function of relative depth. **(b)** Sink rate $\alpha_{j,1}$ (x-axis) against output norm $\|\mathbf{o}_j\|$ (y-axis) for all tokens in 32 batches, for a fixed $(l, h)$ choice in LLAMA-3.1-8B. We plot in red the prediction obtained via our model outlined in Equation (20), and colour each token according to its relative position in the sequence. **(c)** Change in perplexity ($\Delta$PPL) as the sink token's value-vector norm is scaled by a factor in $[0, 1]$, showing the mean and 25th–75th percentile band.

($R^2 = 0.079$). This reveals that first-token contributions depend on the magnitude and directional alignment of value vectors rather than attention allocation.

**Tail tokens: attention and alignment jointly matter.** For tail tokens, contribution depends on both attention and value geometry. Attention alone is the dominant factor but still captures less than half the variance ($R^2 = 0.449$), while the joint effect of attention and value vector alignment provides the strongest prediction ($R^2 = 0.834$). This demonstrates that tail token contributions arise when a token receives attention and its value vector aligns with the output direction.

## 6.2. Sink as Information Suppressor.

Having established that sink token contributions are governed by value geometry, we now examine how this geometry functions mechanically. Analyzing the relationship between sink token value norms and their alignment with semantic tokens reveals that, contrary to existing belief, sink tokens operate as active information suppressors.

**Sink tokens are anti-aligned with semantic tokens.** Prior explanations on the role of sink tokens have largely fo-

cused on value magnitudes: since initial value vectors $\tilde{\boldsymbol{v}}_1^{(l,h)}$ have significantly smaller norms than semantic tokens (Guo et al., b), attending to them produces negligible head output,

$$\|\sum_{j=1}^{i} \alpha_{ij}\tilde{\boldsymbol{v}}_j^{(l,h)}\| \approx \|\alpha_{i1}\tilde{\boldsymbol{v}}_1^{(l,h)}\| \approx \|\tilde{\boldsymbol{v}}_1^{(l,h)}\| \approx 0 \quad (19)$$

However, beyond low magnitudes we observe that sink token value vectors exhibit systematic directional opposition to semantic tokens (Figure 5, left). Across layers, initial token values maintain negative cosine similarity with tail values, $\cos(\tilde{\boldsymbol{v}}_1^{(l,h)}, \tilde{\boldsymbol{v}}_j^{(l,h)}) \approx -0.2$ for all $j \neq 1$, while tail tokens exhibit positive mutual alignment $\cos(\tilde{\boldsymbol{v}}_i^{(l,h)}, \tilde{\boldsymbol{v}}_j^{(l,h)}) \approx 0.2$ for $i, j \neq 1$. This anti-alignment pattern is unexplained by existing theories and, as we show in Table 1, plays an important role in shaping sink token contributions.

**Sink tokens actively oppose semantic tokens.** This raises a key question, *does this directional opposition serve a functional role?* To test this, we perform an intervention, scaling the norm of the initial token's value vector to 0 in layers exhibiting attention sink (2–30 for Llama-8B):

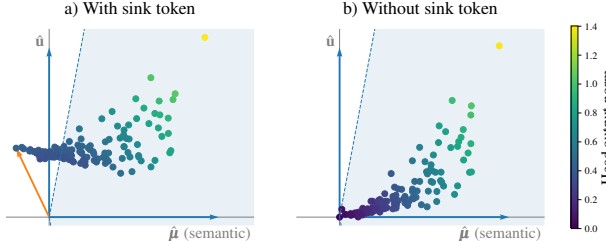

*Figure 6.* **Sinks oppose small magnitude attention weights.** Head outputs $o_i^{(l,h)}$ projected onto the semantic direction $\hat{\boldsymbol{\mu}}$ and its orthogonal complement $\hat{\mathbf{u}}$, coloured by output norm. The dashed cone contains 90% of non-sink values; the orange arrow shows the sink vector. (a) With the sink token present, low-norm heads (dark blue) lie outside the semantic cone, either orthogonal or negatively aligned with $\hat{\boldsymbol{\mu}}$. (b) When the sink value norm is set to zero, the same low-norm heads collapse into the semantic cone. This demonstrates that sink tokens actively prevent weak attention from introducing noisy semantic drift.

$\hat{\boldsymbol{v}}_1^{(l,h)} = \gamma \boldsymbol{v}_1^{(l,h)}$ for $0 < \gamma < 1$. Results are plotted in Figure 6c for Llama-8B, averaging over 128 FineWeb-Edu sequences of 1024 tokens. If sink tokens were simply passive placeholders designed to absorb attention without contributing meaningfully, reducing their norm should minimally affect model performance. Instead, perplexity increases monotonically as $\gamma$ decreases, indicating that initial tokens play an active role: by opposing the semantic direction of non-sink tokens, they exert meaningful influence on model predictions despite their small magnitudes.

**Sink suppression exhibits convex geometry, not monotonic decay.** To understand the purpose of the sink's semantic opposition, we examine it's effect on the head output norm. Prior work assumes that sink value vectors $\tilde{\boldsymbol{v}}_1^{(l,h)}$ exhibiting significantly lower norms than semantic tokens (Guo et al., b) renders the head inactive (Guo et al., a). Under this perspective, one would assume a monotonic suppression mechanism where output norm scales with sink attention $\alpha_{i1}$, and the maximal suppression (minimum output norm) would occur strictly when the sink captures all attention ($\alpha_{i1} \to 1$).

We challenge this intuition, demonstrating that the relationship between sink rate and output norm is fundamentally *convex*, not monotonic. As shown in Figure 5b, the minimum output norm is empirically achieved at sink rates significantly below 1. We formalize this in Appendix E using a geometric model that approximates the output norm as the root of a quadratic:

$$||\mathbf{o}_i|| \simeq \sqrt{\alpha^2 ||\tilde{\boldsymbol{v}}_1||^2 + 2\alpha(1-\alpha)\gamma + (1-\alpha)^2 \kappa^2} \quad (20)$$

where $\alpha$ denotes the sink rate, $\kappa$ captures the average norm $\mathbb{E}_{j>1}[||\tilde{\boldsymbol{v}}_j||]$ of non-sink values, and $\gamma$ their aggregate alignment $\mathbb{E}_{j>1}[\langle \tilde{\boldsymbol{v}}_1, \tilde{\boldsymbol{v}}_j \rangle]$ with the sink token.

Crucially, the location of the minimum depends on $\gamma$. If

sink tokens were strongly aligned with semantic content ($\gamma \gg 0$), suppression would indeed be monotonic in attention offloading. However, we find that sink vectors are consistently *anti-aligned* with other tokens ($\gamma < 0$, see Figure 5a). This directional opposition creates destructive interference, causing the output norm to collapse to its minimum at an intermediate attention threshold $\alpha_{\min} < 1$. These findings reveal that attention sinks function as information suppressors not just through low magnitude, but through a delicate balance of value alignment and attention head characteristics.

**Sink tokens counteract Semantic Drift from weak attention.** We hypothesize that directional opposition between sink and non-sink tokens serves to nullify semantic drift introduced by many small but non-zero attention weights. For a given query, numerous attention weights $\alpha_{ij}$ are small yet non-negligible, and because non-sink tokens align along a shared semantic direction, their weak positive contributions accumulate and bias the attention output.

To show how the sink tokens achieve we this we can use a simple geometric model. Let us define $\hat{\boldsymbol{\mu}}$ as the unit semantic direction defined as the mean of non-sink token value vectors $\hat{\boldsymbol{\mu}} = \frac{1}{T-1}\sum_{j=2}^{T} \tilde{\boldsymbol{v}}_j$. We write the sink value vector as $\boldsymbol{s} = a\hat{\boldsymbol{\mu}} + b\hat{\boldsymbol{u}}$, where $\hat{\boldsymbol{u}}$ is orthogonal to $\hat{\boldsymbol{\mu}}$ and $a, b \in \mathbb{R}$. This decomposition reveals two complementary roles. The *cancellation* role is governed by $a = \boldsymbol{s} \cdot \hat{\mu} < 0$: this negative component subtracts from the positive drift along $\hat{\mu}$ induced by many small attention weights. The *preservation* role is governed by the orthogonal component $b$, which prevents collapse into the semantic cone.

Figure 6 illustrates this geometry empirically, plotting the sink vector $\tilde{\boldsymbol{v}}_1^{(l,h)}$ and the projections of the output $\boldsymbol{o}_i^{(l,h)}$ of a given head onto the plane spanned by the semantic direction $\hat{\boldsymbol{\mu}}$ and by $\hat{\boldsymbol{u}}$. The shaded region in each plot corresponds to the semantic cone, which we define as the region around $\hat{\boldsymbol{\mu}}$ in which 90% of the value vectors lie within. Figure 6a shows that with the sink token present, heads with small output norm lie outside of the semantic cone and are either slightly negatively aligned or orthogonal to the semantic direction. Figure 6b however shows that if we intervene and set the norm of the sink value vector to 0, negating its cancellation effect, then all of the small norm head outputs lie within the semantic cone. This provides evidence that without the sink tokens uninformative value vectors can contribute noisily along the semantic axis.

# 7. Conclusion

We introduced Contribution Weights, a geometry-aware metric that accounts for value vector norms and alignment. Through intervention analysis, we showed that contribution weights provide a more faithful measure of token functional

importance, consistently outperforming existing attention and norm-based approaches across the decoder-only models we evaluate. Decomposing contribution weights into attention, relative norm, and directional alignment revealed the critical role of value geometry in shaping token influence. Our analysis uncovered a key mechanistic insight: sink tokens actively suppress information through directional opposition to semantic tokens, counteracting drift from weak attention accumulation via convex geometric relationships rather than simple monotonic suppression. These findings demonstrate that analyzing value geometry is essential for faithful mechanistic understanding of decoder-only transformers. While our experiments focus on this setting, contribution weights depend only on the structure of the attention mechanism and are in principle applicable to any attention-based architecture such as encoder-only models and ViTs.

## Impact Statement

This paper presents work whose goal is to advance the field of Machine Learning. There are many potential societal consequences of our work, none which we feel must be specifically highlighted here.

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

## A. Notation

Throughout this work we denote matrices with bold upper-case letters (e.g. $\mathbf{A}$, $\mathbf{B}$) and vectors with bold lower-case letters (e.g. $\boldsymbol{a}_i$, $\boldsymbol{b}_i$), using the same letter of the alphabet to show the rows of a matrix (e.g. $\boldsymbol{a}_i$ is the $i$-th row of $\mathbf{A}$) and italic upper-case letters for learnable parameter matrices (e.g. $\boldsymbol{W}_O$).

## B. Limitations of Existing Contribution Measures

We now examine how existing approaches to measuring token contributions often overlook or simplify the three interacting effects identified above. We analyze these limitations using empirical data from LLaMA-3.1-8B, evaluated on Fineweb-Edu.

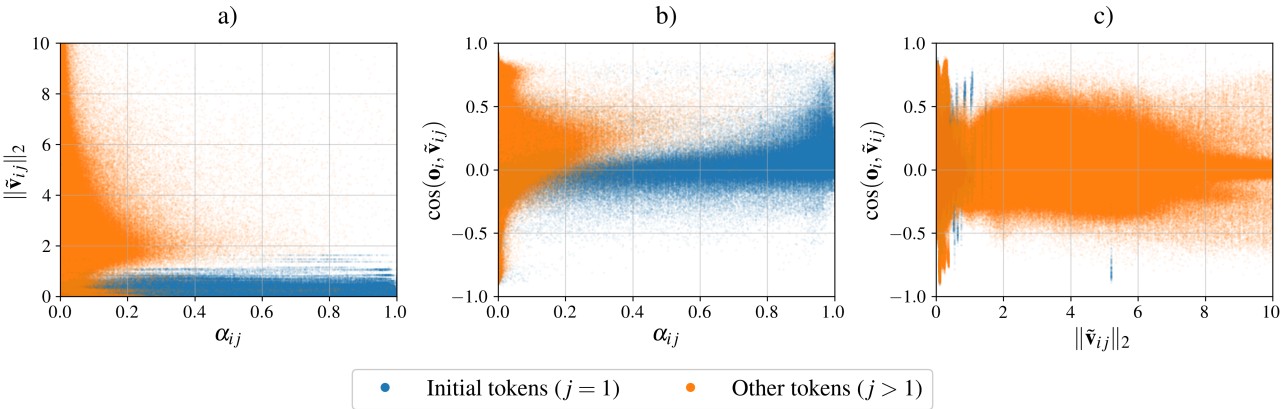

*Figure 7.* **Variation of attention weights, value norms and value alignment.** Pairwise relationships between attention weights $\alpha_{ij}^{(l,h)}$, projected value norms $\|\tilde{\boldsymbol{v}}_j^{(l,h)}\|_2$ and alignment with the output direction $\cos(\boldsymbol{o}_i^{(l)}, \tilde{\boldsymbol{v}}_j^{(l,h)})$ across all layers $l$ and heads $h$ of Llama-8B. The large dispersion across all three plots shows how these factors vary independently. Indeed, high attention weights do not guarantee high large norms or positive alignment, violating the assumptions of weight and norm-based contribution analysis.

**Weight-Based Analysis.** Analyzing token contributions based solely on attention weights $\alpha_{ij}$ implicitly assumes that: (1) projected value norms are approximately constant $\|\tilde{\boldsymbol{v}}_i\|_2 \approx$ const, and (2) all projected value vectors are similarly aligned with the output, $\cos(\boldsymbol{o}_i, \tilde{\boldsymbol{v}}_j) \approx 1$. Under these assumptions, all variation in contribution is explained entirely by differences in attention weights. In reality, however, these assumptions are strongly violated. Figure 7 shows the joint distributions of attention weights $\alpha_{ij}^{(l,h)}$, projected value norms $\|\tilde{\boldsymbol{v}}_j^{(l,h)}\|_2$, and cosine similarities $\cos(\boldsymbol{o}_i^{(l,h)}, \tilde{\boldsymbol{v}}_j^{(l,h)})$ for randomly sampled token pairs. Even when $\alpha_{ij}$ is fixed, both $\|\tilde{\boldsymbol{v}}_j\|_2$ and $\cos(\boldsymbol{o}_i, \tilde{\boldsymbol{v}}_j)$ exhibit substantial variation, particularly for lower attention weights ($\alpha_{ij} < 0.2$).

**Norm-Based Analysis.** Value-weighted attention (Kobayashi et al., a;b) extends weight-based analysis by scaling attention weights by the norm of the projected value vector, $\alpha_{ij}\|\tilde{\boldsymbol{v}}_j\|$. While this accounts for magnitude differences between tokens, it still ignores directional alignment. As shown in Figure 7b and 7c, cosine similarity between projected values and output vectors has a distribution approximately centered at zero (mean $\approx 0.012$). This indicates that value vectors are roughly isotropically oriented relative to the output direction, meaning their contributions can constructively or destructively interfere depending on their relative orientations. Such interference effects cannot be captured by magnitude alone.

**Distance-Based Analysis.** To account for differing directions Ferrando et al. (a) propose a distance based metric that measures the contribution by the $\ell_1$-distance between the projected value and the output $d_{ij} = \|\boldsymbol{o}_i - \boldsymbol{v}_j \boldsymbol{W}_O\|_1$. However, distance metrics are insensitive to the nature of interference and cannot distinguish constructive from destructive contributions. Further, distance metrics do not decompose the output linearly, offering no principled way to normalize distances $d_{ij}$ into contribution weights that sum to a meaningful total, such as the output norm or energy.

## C. Extended Token Importance Results

Here we ablate our analysis from Section 5 by exploring three alternative intervention settings:

1. *Dataset.* We change the dataset to intervene on FineWeb-Edu.

2. *Computation Graph.* We compute and apply *global percentiles* rather than layer-specific thresholds.

3. *Sink.* We permit the removal of semantically void sink tokens.

### C.1. Dataset

We first examine whether our findings generalize across domains by evaluating on FineWeb-Edu (Figure 8). The hierarchy of improvement observed on GSM8K remains consistent: contribution weights (blue) provide the most reliable signal for identifying both critical and dispensable tokens. This confirms that our metric's efficacy is robust to domain shifts and not an artifact of the specific reasoning patterns found in mathematical datasets. The only notable deviation occurs in DeepSeek-7B when removing high-importance tokens, where attention exhibits a more pronounced concave profile than in GSM8K, though it rapidly converges to the relative ranking observed in the original experiment.

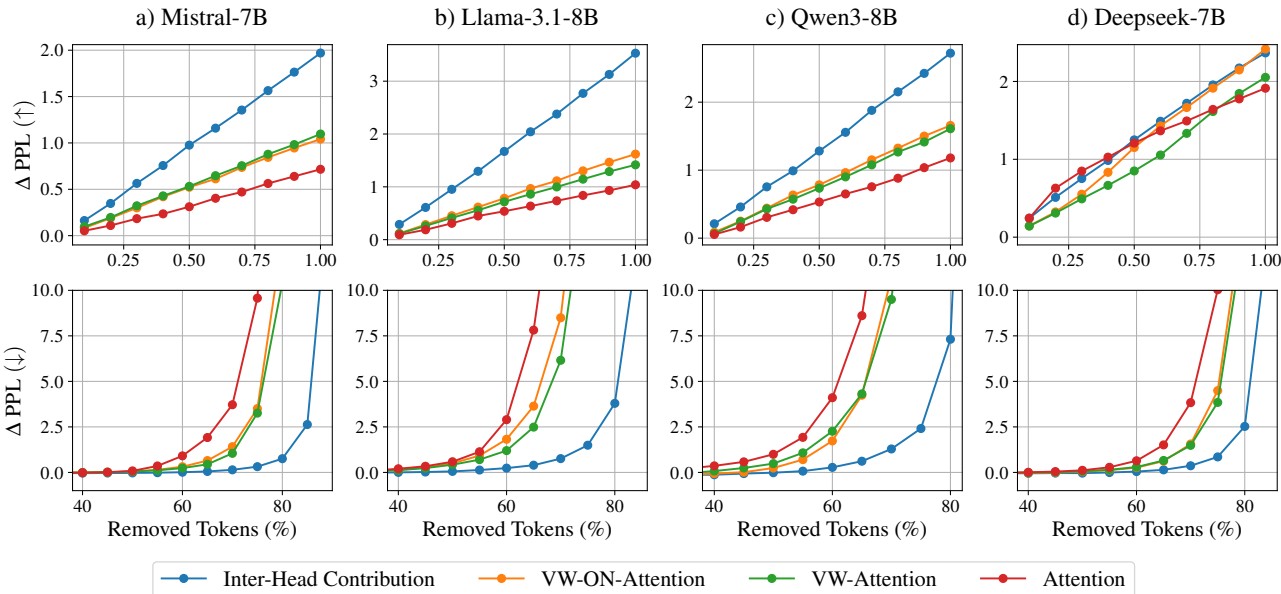

*Figure 8.* **Intervention dataset ablation.** Language modeling loss when removing tokens based on different importance metrics on FineWeb-Edu. Top row shows removal of high-importance tokens; bottom row shows removal of low-importance tokens. Contribution weights (blue) consistently outperform attention-based baselines (red, orange) across all models, demonstrating robustness to domain shifts from mathematical reasoning (GSM8K) to general web text.

### C.2. Computation Graph

We next test whether layer-specific normalization is necessary by computing importance scores using *global percentiles* (Figure 9). This setting implicitly assumes layer contributions are directly comparable across network depths. However, prior work has established that this assumption often does not hold, as later layers can be significantly less influential than preceding ones (Gromov et al., 2025). Selecting tokens based off of their global importance disrupts pruning stability, particularly for models exhibiting multiple sinks such as Qwen3-8B and DeepSeek-7B, where results appear more volatile. Despite these structural instabilities, contribution weights consistently maintain their advantage over attention-based baselines across the majority of conditions.

### C.3. Sink Tokens

Finally, we examine the pathological behavior induced when attention sinks are not protected from removal (Figure 10). When removing *high-importance* tokens (top row), attention-based metrics (red) induce immediate model collapse. This

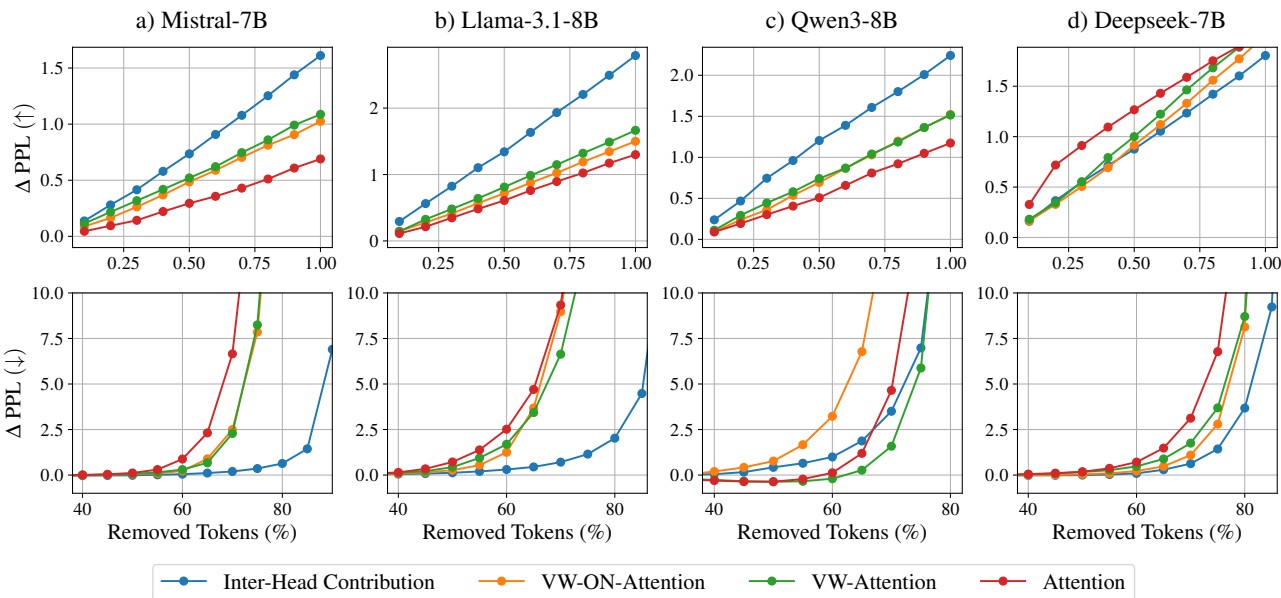

*Figure 9.* **Computation graph ablation using global percentiles.** Language modeling loss when importance scores are computed using global percentiles across all layers rather than layer-specific thresholds. While this introduces instability in models with multiple sinks (Qwen3-8B, DeepSeek-7B), contribution weights (blue) maintain their advantage over attention-based methods, though the gap narrows compared to layer-specific normalization.

confirms that attention heavily overfits to structural sinks, erroneously identifying them as the most critical tokens; their removal destroys the model's functional integrity. In contrast, contribution weights geometrically demote these sink tokens, prioritizing semantic content and yielding a significantly smoother degradation curve. In the *low-importance* removal regime (bottom row), Qwen3-8B emerges as an outlier. This deviation is attributable to the model's reliance on multiple attention sinks, which are classified as marginal by most metrics, leading to their early removal and a consequent drop in performance not seen in single-sink architectures.

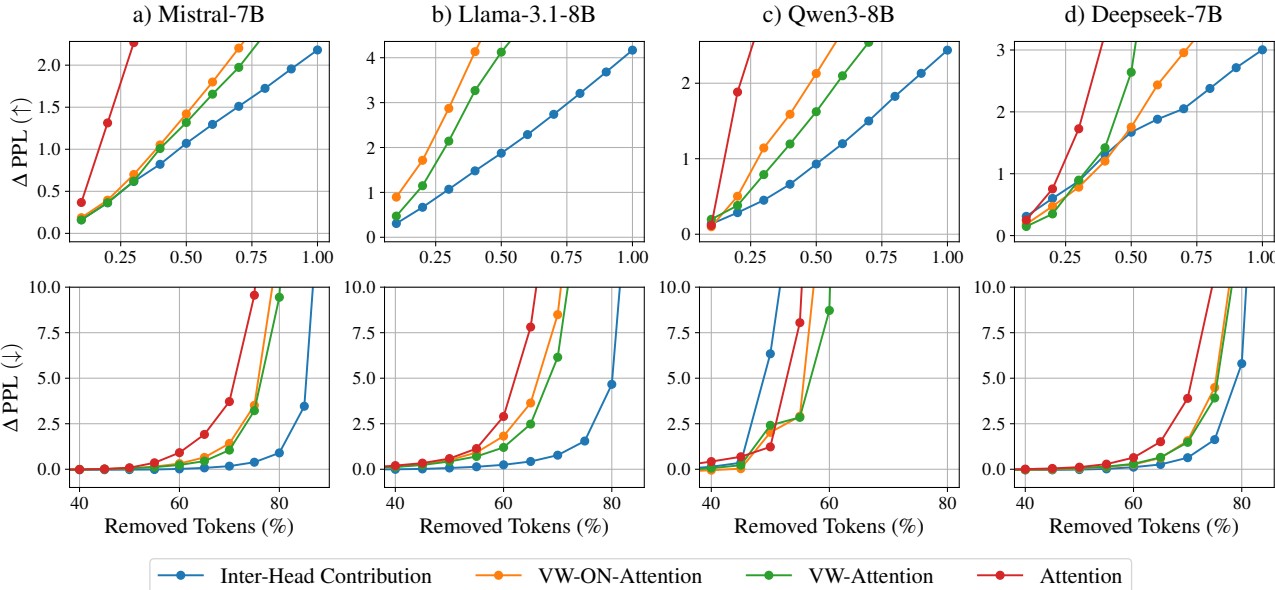

*Figure 10.* **Sink token ablation.** Language modeling loss when attention sinks are permitted to be removed. Top row (high-importance removal): attention-based metrics (red) cause immediate collapse by removing structural sinks, while contribution weights (blue) prioritize semantic content. Bottom row (low-importance removal): Qwen3-8B shows early degradation due to multiple sinks being classified as low-importance and removed, while single-sink architectures remain stable.

# D. Extended Results on the Functional Role of Value Geometry

Here we provide extended results relative to Section 6.1.

## D.1. Contribution components are weakly correlated.

We quantify the independence of each component of the contribution weight decomposition by computing pairwise Pearson correlations between the log-transformed components,

$$\log|c_{ij}| = \log \alpha_{ij} + \log\left(\frac{\|\tilde{\boldsymbol{v}}_j\|}{\|\boldsymbol{o}_i\|}\right) + \log\left|\cos(\boldsymbol{o}_i, \tilde{\boldsymbol{v}}_j)\right|, \tag{21}$$

The log transformation converts the multiplicative form into an additive one, enabling meaningful assessment of linear dependence. Because contribution weights can be signed, all analyses are performed on their magnitudes. Table 2 reports pairwise correlations for LLaMA-3.1-8B for 32 different sequences of length 128 across all token positions, heads and layers. Attention is negatively correlated with relative norm ($r = -0.287$), affirming our previous observations that when attention is large value norm is small and vice versa, and weakly correlated with cosine similarity ($r = 0.147$). Cosine similarity and relative norm are uncorrleated ($r = -0.07$). Overall, these low correlations suggest that each component captures a distinct factor of variation, and that high attention alone does not imply high contribution—its effect depends on whether value geometry (norms and alignment) amplifies or attenuates the attention signal.

*Table 2.* **Correlation of contribution components.** Pairwise Pearson correlations between log-transformed contribution components for LLaMA-3.1-8B. Correlations are computed across 32 sequences of length 128, aggregating over all token positions, attention heads, and layers. Low correlations indicate that attention weight, relative norm, and cosine similarity capture distinct aspects of token contribution.

| | Correlation | | |
| **Model** | Attention | Relative Norm | Cosine Sim. |
| --- | --- | --- | --- |
| Attention | 1.000 | -0.287 | 0.147 |
| Relative Norm | -0.287 | 1.000 | -0.070 |
| Cosine Sim. | 0.147 | -0.070 | 1.000 |

## D.2. Variance Decomposition Experimental Setup.

We evaluate three token groups: (i) all tokens $j \in \{1, \ldots, T\}$, (ii) the initial token $j = 1$, and (iii) non-initial tokens $j > 1$. We compute the contribution weights and their components for three models: LLaMA-3.1-8B, using two datasets: FineWeb-Edu and GSM8K, with 32 samples from each and a maximum sequence length of 512. For each regression we uniformly sample 5 million data points performing 5 fold cross validation, evaluating on a held-out test set. Model-wise $R^2$ results for LLaMA-3.1-8B are reported in Table 3.

## D.3. Extended Results on GSM8K

Below we provide further results, following the same experimental setup as described above but using GSM8K instead of FineWeb-Edu. The $R^2$ values are very similar across datasets, suggesting that the importance of each component is largely indpendent of the data source.

*Table 3.* **Regression analysis of contribution weight components.** We report the coefficient of determination ($R^2$) for linear regressions of contribution weights onto individual and combined components. Each row corresponds to a different regression model: (1) the scalar attention weight $\log(\alpha_{ij})$, (2) the relative norm of the projected value vector $\text{norm}_{ij} = \log(\|\tilde{\boldsymbol{v}}_j\|_2/\|\boldsymbol{o}_i\|_2)$, (3) the cosine similarity $\log|\cos(\tilde{\boldsymbol{v}}_j, \boldsymbol{o}_i)|$, and (4–6) combinations of these components. We evaluate regressions under three token groupings: (i) $j \in \{1, T\}$ (all tokens), (ii) $j = 1$ (initial tokens), and (iii) $j > 1$ (non-initial tokens). Higher $R^2$ values indicate that the corresponding component or combination explains more of the contribution weight variance.

| | FineWeb-Edu ($R^2$) | | | GSM8K ($R^2$) | | |
| **Components** | $j \in \{1, T\}$ | $j = 1$ | $j > 1$ | $j \in \{1, T\}$ | $j = 1$ | $j > 1$ |
|---|---|---|---|---|---|---|
| $\alpha_{ij}$ | 0.055 | 0.079 | 0.449 | 0.060 | 0.085 | 0.455 |
| $\frac{\|\tilde{\boldsymbol{v}}_j\|}{\|\boldsymbol{o}_i\|}$ | 0.000 | 0.027 | 0.000 | 0.000 | 0.022 | 0.000 |
| $\cos(\boldsymbol{o}_i, \tilde{\boldsymbol{v}}_j)$ | 0.036 | 0.518 | 0.028 | 0.034 | 0.496 | 0.028 |
| $\alpha_{ij}, \frac{\|\tilde{\boldsymbol{v}}_j\|}{\|\boldsymbol{o}_i\|}$ | **0.489** | 0.157 | 0.579 | **0.511** | 0.145 | 0.587 |
| $\alpha_{ij}, \cos(\boldsymbol{o}_i, \tilde{\boldsymbol{v}}_j)$ | 0.305 | 0.611 | **0.834** | 0.303 | 0.602 | **0.863** |
| $\frac{\|\tilde{\boldsymbol{v}}_j\|}{\|\boldsymbol{o}_i\|}, \cos(\boldsymbol{o}_i, \tilde{\boldsymbol{v}}_j)$ | 0.020 | **0.725** | 0.021 | 0.026 | **0.689** | 0.026 |

## D.4. Value geometry suppresses high attention.

Initial tokens receive disproportionate attention mass but contribute far less than their attention weights suggest.

We quantify this via the average first-token inter-head contribution $\mathbb{E}c_1$, defined, for any given input sequence, as the mean first token contribution across all layers, heads, and query positions:

$$\mathbb{E}[\hat{c}_1^{(l)}] = \frac{1}{T}\sum_{h=1}^{H}\sum_{i=1}^{T}\hat{c}_{i1}^{(l,h)}, \qquad \mathbb{E}[\hat{c}_1] = \frac{1}{L}\sum_{l=1}^{L}\mathbb{E}[\hat{c}_1^{(l)}] \tag{22}$$

where the layer-wise average $\mathbb{E}\hat{c}^{(l)}$ sums over heads such that contribution rows are normalized[2]. Table 4 compares initial-token weights for LLaMA models on FineWeb-Edu across three metrics: attention, value-weighted attention, and inter-head contribution.

*Table 4.* **Average weight on the first token.** Average weight on the first token (Equation (22)) for LLaMA models on FineWeb-Edu across three metrics: 1) attention, 2) value-weighted attention, and 3) contribution. Value-weighted (VW) and contribution weights assign substantially less mass to the first token than attention, indicating that value magnitude and direction mitigate the overemphasis introduced by softmax normalisation.

| | Average Weight on First Token | | |
| **Model** | Attention | VW-Attention | Contribution |
|---|---|---|---|
| LLaMA-1B | 0.632 | 0.147 | 0.243 |
| LLaMA-3B | 0.708 | 0.159 | 0.142 |
| LLaMA-8B | 0.708 | 0.159 | 0.142 |

Across all models, attention consistently overestimates first-token importance. Value-weighted attention (which incorporates value magnitude) reduces this overemphasis, while contribution weights (incorporating both value magnitude and directional alignment) reduce it further still. In LLaMA-3.1-8B, for instance, attention assigns $\hat{\alpha}_1 = 0.708$ to the first token, yet it contributes only $\hat{c}_1 = 0.142$, corresponding to a $5\times$ suppression. Section 6.1 established that value geometry largely explains first-token contributions, with the product of relative value norm and cosine similarity accounting for $R^2 = 0.725$ of variance. We now examine how these geometric factors jointly suppress first-token influence despite high attention, exploring their functional role in the attention mechanism.

---

[2]For attention and value-weighted attention, which are not normalized over heads, we compute layer-wise averages as $\mathbb{E}\alpha_1^{(l)} = \frac{1}{HT}\sum_{h=1}^{H}\sum_{i=1}^{T}\alpha_{i1}^{(l,h)}$ to maintain comparable scales.

## D.5. Value geometry concentrates contributions.

For tail tokens ($j > 1$), value geometry concentrates rather than suppresses contributions. We quantify concentration via entropy $S^{(l,h)}$ for each head:

$$\bar{c}_{ij}^{(l,h)} = \frac{|c_{ij}^{(l,h)}|}{\sum_{k=1}^{i} |c_{ik}^{(l,h)}|}, \qquad S^{(l,h)} = \frac{1}{T} \sum_{i=1}^{T} \sum_{j=1}^{i} -\bar{c}_{ij}^{(l,h)} \log \bar{c}_{ij}^{(l,h)} \tag{23}$$

where normalization accounts for signed contributions and lower entropy indicates greater concentration. Table 5 reports average entropy for LLaMA models, including and excluding the first token. The attention sink strongly affects apparent attention concentration: for LLaMA-8B, attention entropy is $S = 1.21$ including the first token (appearing highly concentrated due to the sink), but rises to $S = 2.61$ when excluding it (revealing the true spread across tail tokens). Comparing tail tokens only, value-weighted attention has $S = 2.58$ and contribution weights have $S = 2.27$, both lower than attention's $S = 2.61$. The entropy reduction from both attention metrics to contribution demonstrates that the cosine similarity between tokens and the output helps to concentrate attention: tokens with high attention but poor value-output alignment contribute less than their attention weights indicate, while geometrically aligned tokens contribute more, resulting in a more concentrated effective distribution.

*Table 5.* **Average entropy of weight matrices.** Average entropy $S$ of: 1) attention, 2) value-weighted attention, and 3) contribution weights (Equation (23)) across layers and heads of LLaMA models on FineWeb-Edu. Entropy is reported for all tokens and with the first token excluded. Including the first token lowers entropy substantially due to its disproportionate attention mass. When excluded, value-weighted attention and contribution weights—which account for value magnitude and direction—show lower entropy, indicating that value information sharpens token selectivity.

| Model | Average Entropy | | | Average Entropy ($j \neq 1$) | | |
|---|---|---|---|---|---|---|
| | Attention | VW-Attention | Contribution | Attention | VW-Attention | Contribution |
| LLaMA-1B | 1.40 | 2.15 | 1.71 | 2.42 | 2.39 | 2.14 |
| LLaMA-3B | 1.21 | 2.22 | 1.72 | 2.54 | 2.51 | 2.20 |
| LLaMA-8B | 1.21 | 2.30 | 1.79 | 2.61 | 2.58 | 2.27 |

## D.6. Interference refines contributions.

We hypothesize that this sharpening is a result of geometric filtering, where the alignment of the attended value vector with the layer's output direction modulates the final contribution. To quantify this effect we consider the relationship between a token's contribution-to-attention ratio,

$$\frac{c_{ij}^{(l,h)}}{\alpha_{ij}^{(l,h)}} = \frac{\|\tilde{\boldsymbol{v}}_j^{(l,h)}\|}{\|\boldsymbol{o}_i^{(l,h)}\|} \cos(\boldsymbol{o}_i^{(l,h)}, \tilde{\boldsymbol{v}}_j^{(l,h)}) \tag{24}$$

and it's cosine similarity with the output $\cos(\boldsymbol{o}_i^{(l,h)}, \tilde{\boldsymbol{v}}_j^{(l,h)})$, making sure to compute contribution with respect to the head-level output $\boldsymbol{o}_i^{(l,h)}$ to avoid interference between heads. The contribution-to-attention ratio isolates the effect of value geometry and how it can boost (positive ratio) or suppress (negative ratio) a token's contribution relative to it's attention weight. Fitting the contribution-to-attention ratio against cosine similarity we observe a very strong near-linear relationship ($r = 0.949$, $p = 0.000$) where: the value vectors are positively aligned contributions are amplified, near-orthogonal vectors are filtered regardless of their attention mass and negatively aligned vectors actively induce destructive interference in the output.

# E. A simple model for the relationship between Sink-Rate and Output Norms

Recall the definition of output as sum of projected value vectors (where we omit layer and head in the notation):

$$\mathbf{o}_i = \alpha_{i,0}\tilde{v}_0 + \sum_{1 \leq j \leq i} \alpha_{i,j}\tilde{v}_j$$

and define the vector $\mathbf{w}_i := \sum_{1 \leq j \leq i} \alpha_{i,j}\tilde{v}_j = \mathbf{o}_i - \alpha_{i,0}\tilde{v}_0$ so that, omitting indices for $\alpha$,

$$
\begin{aligned}
||\mathbf{o}_i||^2 &= ||\alpha\tilde{v}_0 + \mathbf{w}_i||^2 \\
&= \alpha^2||\tilde{v}_0||^2 + 2\alpha\langle\tilde{v}_0, \mathbf{w}_i\rangle + ||\mathbf{w}_i||^2
\end{aligned}
\tag{25}
$$

Figure 11[bottom] shows empirically for a specific choice of $(l, h)$ head how, noticing how at $\alpha = 1$ the variance is null and the resulting value is always 0, both

$$||\mathbf{w}_i|| \simeq \kappa(1 - \alpha), \quad \kappa > 0 \tag{26}$$

and

$$\langle\tilde{v}_0, \mathbf{w}_i\rangle \simeq \gamma(1 - \alpha), \quad \gamma < 0 \tag{27}$$

so that

$$||\mathbf{o}_i||^2 \simeq \alpha^2||\tilde{v}_0||^2 + 2\alpha(\alpha - 1)\gamma + (1 - \alpha)^2\kappa^2 \tag{28}$$

and thus

$$||\mathbf{o}_i|| \simeq \sqrt{\alpha^2||\tilde{v}_0||^2 + 2\alpha(1 - \alpha)\gamma + (1 - \alpha)^2\kappa^2} \tag{29}$$

In Figure 11[top], we present the distribution of $R^2$ scores across heads for each layer. The consistently high $R^2$ values (median $> 0.8$ and $> 0.7$ across most layers) indicate that the linear model captures the relationship with high fidelity.

In an attempt to explain this phenomenon, recall how

$$\mathbf{w}_i := \sum_{1 \leq j \leq i} \alpha_{i,j}\tilde{v}_j$$

and since the constraint on attention weight is $1 = \alpha_{i,0} + \sum_{1 \leq j \leq i} \alpha_{i,j}$ it must hold that

$$\sum_{1 \leq j \leq i} \alpha_{i,j} = 1 - \alpha_{i,0}$$

From these observations it follows that if we define $p_{i,j} := \frac{\alpha_{i,j}}{1 - \alpha_{i,0}}$ for $1 \leq j \leq i$ then these scalars are positive and sum to 1, indeed they are the renormalized attention distribution once the [BOS] token is removed. Note that with this renormalisation in mind one has

$$\mathbf{w}_i := (1 - \alpha_{i,0}) \sum_{1 \leq j \leq i} p_{i,j}\tilde{v}_j = (1 - \alpha_{i,0})\hat{\mathbf{w}}_i$$

where $\hat{\mathbf{w}}_i := \sum_{1 \leq j \leq i} p_{i,j}\tilde{v}_j = \mathbb{E}_{p_{i,\cdot}}[\tilde{v}]$.

The previous empirical results are telling us that both $||\hat{\mathbf{w}}_i||$ and $\langle\hat{\mathbf{w}}_i, \tilde{v}_0\rangle$ are approximately constant. Since $\hat{\mathbf{w}}_i$ is the expected value under $p_{i,\cdot}$ this roughly follows from other observations on the $\tilde{v}_j$, namely that their norms are of the same magnitude and that their dot product with $\tilde{v}_0$ is constant (and negative).

Note that the approximate equality

$$||\mathbf{o}_i|| \simeq \sqrt{\alpha^2||\tilde{v}_0||^2 + 2\alpha(1 - \alpha)\gamma + (1 - \alpha)^2\kappa^2} \tag{30}$$

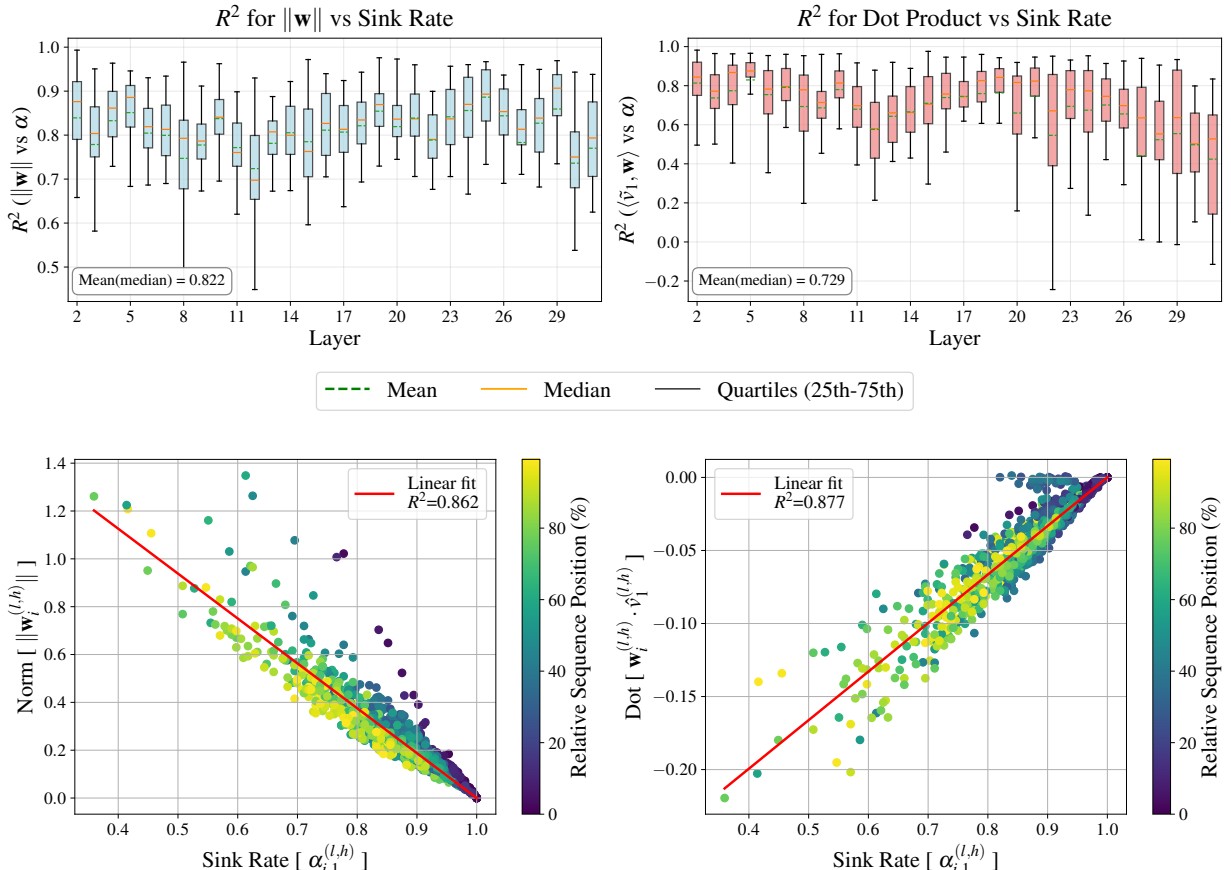

*Figure 11.* Left: Linear fit of $\|\mathbf{w}_i^{(l,h)}\|$ against $\alpha_{i,1}^{(l,h)}$. Right: Linear fit of $\langle \tilde{v}_1^{(l,h)}, \mathbf{w}_i^{(l,h)} \rangle$ against $\alpha_{i,1}^{(l,h)}$. Top: Aggregate $R^2$ statistics. Bottom: Fit for a fixed index $(l,h)$, each point corresponds to a token coloured according to its relative sequence position.

implies that the minimum norm is not attained at $\alpha = 1$ as a simplistic understanding of sink token as suppressor would lead to think, but is expected at the point

$$\alpha = \frac{\kappa^2 - \gamma}{\kappa^2 - 2\gamma + \|\tilde{v}_0\|^2} = \frac{\|\hat{\mathbf{w}}\|^2 - \langle \tilde{v}_0, \hat{\mathbf{w}} \rangle}{\|\hat{\mathbf{w}} - \tilde{v}_0\|^2} \tag{31}$$

In fact the minimum in the interval $[0, 1]$ occurs at $\alpha = 1$ only if

$$\|\hat{\mathbf{w}}\|^2 - \langle \tilde{v}_0, \hat{\mathbf{w}} \rangle \geq \|\hat{\mathbf{w}}\|^2 - 2\langle \tilde{v}_0, \hat{\mathbf{w}} \rangle + \|\tilde{v}_0\|^2 \iff \|\tilde{v}_0\|^2 \leq \langle \tilde{v}_0, \hat{\mathbf{w}} \rangle \iff \frac{\|\tilde{v}_0\|}{\|\hat{\mathbf{w}}\|} \leq \cos(\tilde{v}_0, \hat{\mathbf{w}}),$$

which is empirically never the case as the anti-alignment of the sink value with non-sink values leads to $\cos(\tilde{v}_0, \hat{\mathbf{w}}) < 0$.

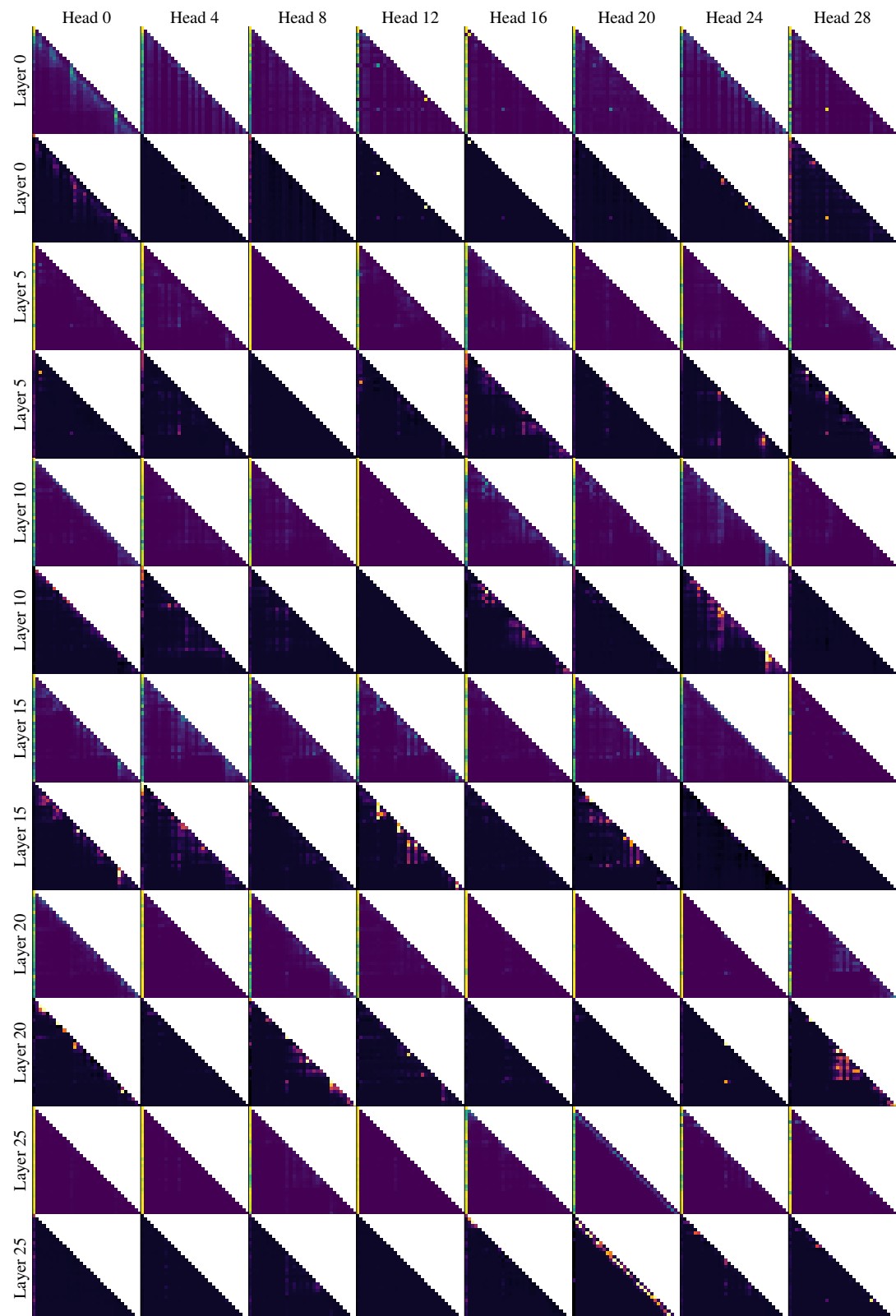

*Figure 12.* **Contribution weight matrices**. Attention and contribution weights computed from LLaMA-3.1-8B evaluated on FineWeb-Edu. Rows alternate between attention weights and contribution weights. All attention weights and contribution weights are plotted with the same colourbar axis from $[0, 1]$ and $[-0.01, 0.15]$ respectively.

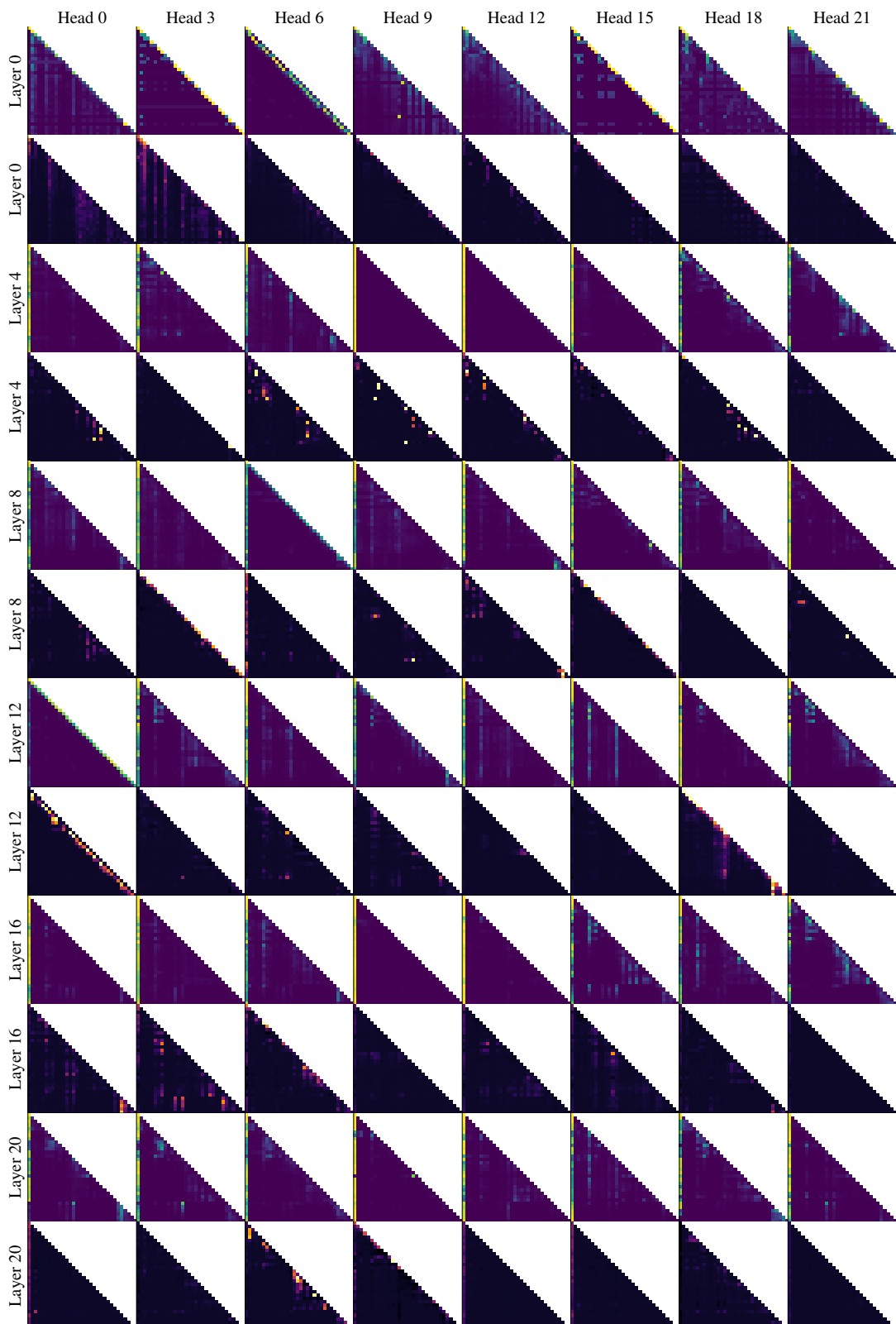

*Figure 13.* **Contribution weight matrices**. Attention and contribution weights computed from Qwen2.5-7B evaluated on FineWeb-Edu. Rows alternate between attention weights and contribution weights. All attention weights and contribution weights are plotted with the same colourbar axis from $[0, 1]$ and $[-0.01, 0.15]$ respectively.

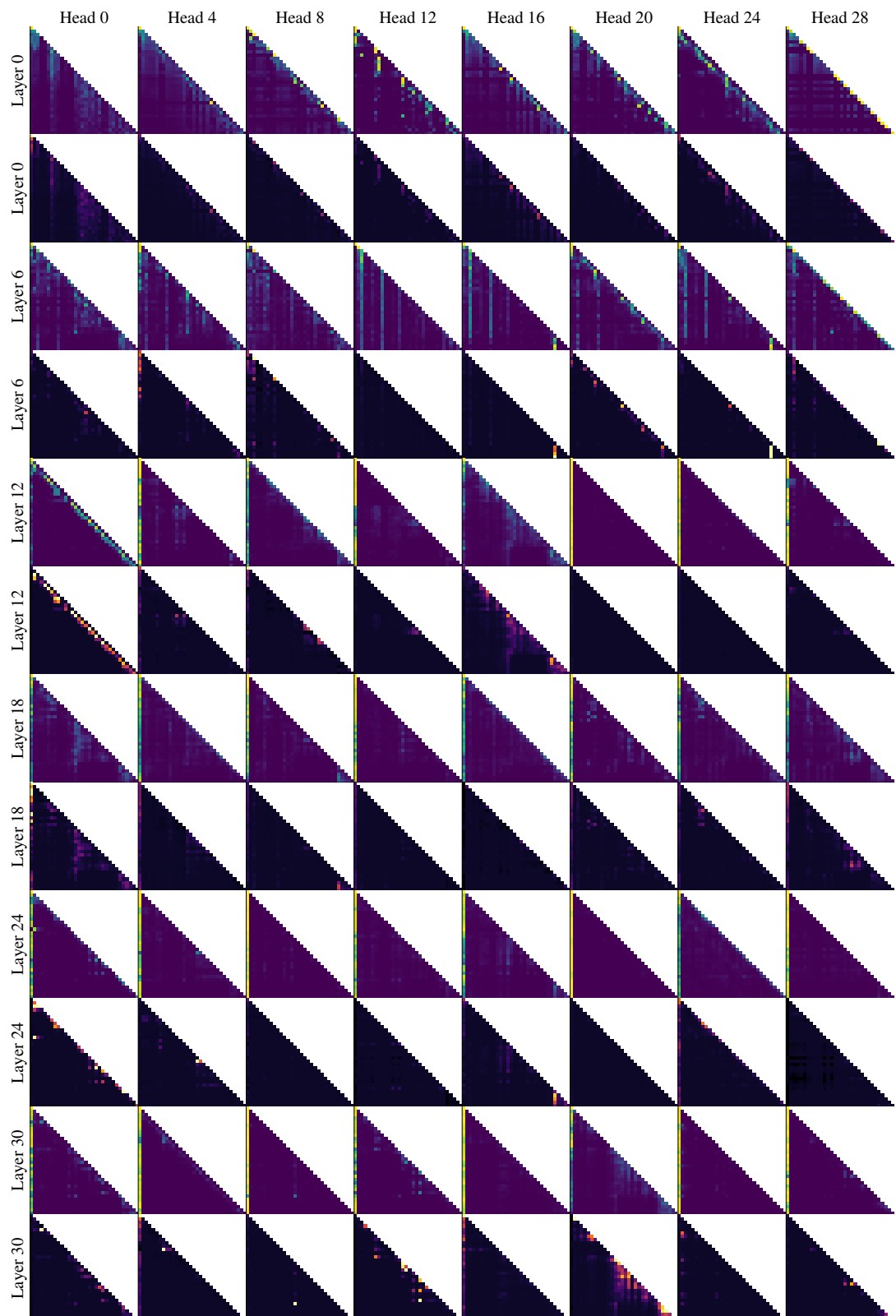

*Figure 14.* **Contribution weight matrices**. Attention and contribution weights computed from Qwen3-8B evaluated on FineWeb-Edu. Rows alternate between attention weights and contribution weights. All attention weights and contribution weights are plotted with the same colourbar axis from $[0, 1]$ and $[-0.01, 0.15]$ respectively.

