# OpenReview forum: "Contribution Weights: A Geometrical Analysis of Self-Attention Transformers"
_ICML.cc/2026/Conference — ICML 2026 regular_

### Official Review · Reviewer_bUeb · 2026-03-11

**Soundness:** 4
**Presentation:** 4
**Significance:** 3
**Originality:** 3
**Overall Recommendation:** 5
**Confidence:** 4

**Summary:**

The authors propose a novel geometry-aware metric, they coin contribution weights, to better measure how much each token actually influences a model's (here, LLMs) output. Their goal is to provide a method that improves interpretability by accounting not only for attention scores but also for the magnitude and directional alignment of value vectors with the output. The main result is showing that interpreting token contribution depends on jointly analyzing the contributions of attention weight, value norm, and value vector cosine alignment with the output. Through experiments on several LLMs and token-removal interventions, they demonstrate that contribution weights more accurately identify semantically important tokens than existing attention-based metrics and provide new insights into attention sinks, showing that these tokens actively suppress information through geometric opposition rather than merely absorbing excess attention. Overall, I am sympathetic with this work. I believe it is a nice addition to the literature, demonstrating the pitfall of certain interpretability methods, providing an elegant, yet simple, view of attention sinks, coupled with thorough analyses.

**Compliance With Llm Reviewing Policy:**

Affirmed.

**Final Justification:**

Solid and well written work. Authors took the time to clarify all my concerns in the rebuttal phase.

**Key Questions For Authors:**

Q1: The authors declare that “Across all models and removal rates, contribution weights consistently outperform attention-based metrics in both comprehensiveness and sufficiency (Figure 3), providing the clearest signal of token functional importance”. Whereas this is true for comprehensiveness, that pattern is less clear for sufficiency (in particular for low removal rates). I would thus suggest the authors to, at the very least, tone down that statement and re-write it such that it aligns better to the data shown in figure 3.

Q2: The paragraph entitled “All geometric information contributes to faithfulness improvements.” is not backed up by any figure. Could the authors relate it to shown data?

Q3: Why are the authors not considering the tri-factor model in their regression analyses displayed in table 1?


Q4: The authors write: “We hypothesize that directional opposition between sink and non-sink tokens serves to nullify semantic drift introduced by many small but non-zero attention weights.” Would it be possible to provide an analysis that compares a corpus of input sequence with larger attention weights compared to another corpus of sentence, and evaluate the role of sink tokens in a more relative manner.

Typos:

Page 6 -> “studying the their significance”
Page 7 -> “Results are plotted in Figure 6c for Llama-8B…” Figure should be 5c.

**Limitations:**

yes

**Strengths And Weaknesses:**

Strengths:

-The paper is remarkably clearly written and structured, making the read very enjoyable.

-The proposed metric is a nice addition to other geometric aware methods

-In particular, the insights derived from the geometric analysis of value vector directionality from sink tokens provides an important piece of the puzzle to understand attention sinks from a novel functional role perspective.

Weaknesses:

-The main weakness of the article holds in not expliciting (throughout the text) what are the added values of their proposed method with respect to previous geometry-aware proposals. For instance, in the related work section, the authors write “existing approaches capture only partial value geometry, neglecting effects such as inter-token interference…”. Such differences should be matched at the formal level to understand better the added value of the proposed contribution.

-The order of the figures and table does not always follow the order of results discussed in the text. Admittedly, this is more of a pet peeve of mine, but I think it's a valid one.

---

> ### Author Rebuttal · Authors · 2026-03-30
>
> We would like to thank the author for taking the time and effort spent understanding our work. We are particularly pleased that they found the geometric analysis of sink tokens **"an important piece of the puzzle"** for understanding attention sinks, and that they considered the paper overall a **"very enjoyable read"**. We hope to address the reviewers points below, and note that the typos on pages 6 and 7 will be corrected in the revision.
>
> ### W1: Added value over existing geometry-aware methods not explicit enough.
> We thank the reviewer for this suggestion. We agree that the formal distinction between contribution weights and prior geometry-aware methods could be made more prominent in the main text. In Appendix B we provide an analysis on the limitation of existing methods, where we show using empirical data from LLaMA-3.1-8B how weight-based analysis ignores variation in value magnitudes, norm-based analysis (Kobayashi et al.) ignores variation in value-output alignment, and distance-based analysis (Ferrando et al.) cannot distinguish constructive from destructive interference (Figure 7). We would be happy to incorporate this discussion into the main body of the paper in the revision to make the formal distinctions clearer.
>
> ### W2: Figure and table ordering.
> Thank you for flagging this. We will reorder figures and tables to follow the order of discussion in the text in the revision.
>
> ### Q1: Sufficiency claim too strong at low removal rates.
> We agree with the reviewer that our statement is stronger than the data warrants given the similarity across measures at low sufficiency removal rates. We propose revising the sentence to "Across all models and at meaningful removal rates, contribution weights consistently outperform attention-based metrics in both comprehensiveness and sufficiency (Figure 3), providing the clearest signal of token functional importance." We note that at higher removal rates, where the distinction between metrics is most practically relevant, contribution weights maintain a clear advantage across all models.
>
> ### Q2: "All geometric information contributes to faithfulness improvements" lacks figure reference.
> Thank you for catching this. The paragraph directly describes the progression visible in Figures 3 and 4 where we see that raw attention performs worst, value-weighted attention improves upon it, VW-ON improves further, and contribution weights (which add angular alignment) perform best. We will add explicit references to Figures 3 and 4 in this paragraph in the revision.
>
> ### Q3: Why is the tri-factor model not included in the regression analysis?
> The regression in Table 1 is in fact regressing onto the contribution weights themselves, which are by definition the product of all three components (Equation 14). Including the full tri-factor product as a regressor would amount to regressing $c_{ij}$ onto itself, trivially yielding $R^2 = 1$. The purpose of Table 1 is to perform an importance analysis over subsets of this decomposition, asking how much of the variance in the total contribution each subset of components captures. The results reveal that cosine similarity alone explains $R^2 = 0.518$ for sink tokens while attention alone explains only $R^2 = 0.079$, and that for non-sink tokens the combination of attention and cosine ($R^2 = 0.834$) substantially outperforms attention and norm ($R^2 = 0.579$). We will clarify this framing in the revision.
>
> ### Q4: Comparative analysis across corpora with different attention weight magnitudes.
> This is an interesting suggestion. Our current analysis in Section 6.2 already examines the relationship between sink rate and output norm across the full distribution of attention weights within each sequence (Figure 5b), and the geometric model in Equation 20 explicitly parameterizes how the suppression effect varies as a function of the sink attention rate $\alpha$. We believe this provides a continuous view of the phenomenon that subsumes a discrete corpus comparison. A between-corpus analysis would additionally need to control for many confounding factors (sequence length, domain, vocabulary distribution) that could obscure the geometric signal. We think the within-sequence interventional approach in Figures 5c and 6, where we directly manipulate sink value norms and observe the causal effect, provides stronger evidence for the active role of sinks than a correlational comparison across corpora would.

---

> > ### Author Rebuttal · Reviewer_bUeb · 2026-04-01
> >
> > I thank the authors for their rebuttal, clarifying the issues I raised, and providing satisfactory answers to my questions. I maintain my score of  5, and congratulate the authors for this nice work.

---

> > > ### Author Response · Authors · 2026-04-04
> > >
> > > We sincerely thank the reviewer for their positive assessment and supportive engagement throughout the review process. We are glad that our rebuttal addressed the raised concerns satisfactorily.

---

### Official Review · Reviewer_KUpN · 2026-03-13

**Soundness:** 3
**Presentation:** 3
**Significance:** 3
**Originality:** 3
**Overall Recommendation:** 4
**Confidence:** 4

**Summary:**

This paper challenges the practice of using attention weight as a metric for token importance in prior work, and proposes Contribution Weight, a more faithful metric. It decomposes Contribution Weight into three components (Attention weights, Relative norm, and Cosine similarity) for in-depth investigation. Experiments prove that Contribution Weight reflects the degree of token importance more faithfully than attention weight. The paper further analyzes the behavior of sink tokens under this framework, and demonstrates that sink tokens actively oppose semantic tokens.

**Compliance With Llm Reviewing Policy:**

Affirmed.

**Final Justification:**

My main concern was the need for some conceptual clarification. The authors have clarified these concepts, so my concerns have been resolved. I have increased my score.

**Key Questions For Authors:**

The current Contribution Weight involves more model modules than attention weight. So I am curious that can acceleration work originally adapted for attention weight be migrated to Contribution Weight?

**Limitations:**

yes

**Strengths And Weaknesses:**

Strengths:
1. The paper has a clear logical chain. It first analyzes the unfaithfulness of the commonly used attention weight in prior work to the actual token importance, then proposes the more faithful Contribution Weight metric and decomposes it into three interpretable components for detailed research. Finally, it uses this framework to analyze the prevalent attention sink phenomenon in attention mechanisms, and draws conclusions that differ from previous studies.
2. The faithfulness of Contribution Weight and attention weight is evaluated via Comprehensiveness and Sufficiency, and the results show that Contribution Weight exhibits more faithful performance.
3. Regression analysis is conducted on the three geometrically meaningful components decomposed from Contribution Weight, verifying that each component makes an indispensable contribution.

Weaknesses:
1. Lack of intuitive case studies. The paper only conducts quantitative research. Although the change in PPL is used as a proxy for the importance of one token to another, PPL cannot fully reflect the real importance, nor can it necessarily translate to performance on practical tasks. Therefore, practical examples should be provided to illustrate the differences in important tokens captured by Contribution Weight and attention weight.
2. In Section 6.2 "Sink tokens actively oppose semantic tokens", first, Figure 6c is missing (it should be Figure 5c). Second, the logic is slightly jumpy. The paper uses "passive" and "active" to describe the function of sink tokens relative to other tokens, but such expressions are vague, which makes the implication of the operation "scaling the norm of the initial token’s value vector to 0" unclear. It is a natural outcome that intervening in the norm of a token causes changes in PPL.
3. The term "geometry" in the paper is somewhat overused. While it is reasonable to use "geometry" to describe the three decomposed metrics of Contribution Weight, the term is applied to many parts involving only basic linear algebra, which is inaccurate.

---

> ### Author Rebuttal · Authors · 2026-03-30
>
> We thank the reviewer for their careful reading and for recognizing the clear logical chain of the paper. We address each of their points below and note that the Figure 6c/5c typo will be corrected in the revision.
>
> ### W1: Lack of intuitive case studies.
> We agree with the reviewer that PPL alone does not fully capture token importance in all settings. This is precisely why we also evaluate faithfulness on downstream tasks in Figure 4, measuring accuracy on MMLU, BoolQ, Lambada, and Winogrande under increasing token removal rates. **The results on these practical tasks are consistent with our PPL findings** that contribution weights maintain baseline task performance at higher removal rates than all other metrics. We believe this provides the practical grounding the reviewer is looking for, but we are open to including additional qualitative case studies in the revision to complement the quantitative results.
>
> ### W2: Passive vs active role of sink tokens, and the intervention result.
> We hope that we can clarify what we mean by passive and active. Prior work has characterized sink tokens as having essentially no functional effect on other tokens. They have been described as "implicit bias terms" (Sun et al.; Darcet et al.) that absorb excess attention and "deactivate heads by driving output norms toward zero" (Bondarenko et al.; Guo et al.), as discussed in Section 2. In other words, the prevailing view is that sink tokens are doing nothing useful, they just soak up spare attention and their small value norms mean they have no real effect on the output. **If that were true, reducing their value norms further should not matter.** Our intervention in Figure 5c tests exactly this and shows the opposite that **perplexity increases steadily as we scale down the sink value norm**, which means sink tokens are actively shaping the output. By active, we mean that sink tokens **actively oppose the semantic direction of non-sink tokens**, suppressing noisy drift that would otherwise accumulate from many small attention weights. Figure 5a shows this directional opposition directly, and Figure 6 shows that without sinks, low-norm head outputs collapse into the semantic cone and introduce noisy drift. **Under the passive view of sinks, none of this should happen.**
>
> ### W3: Overuse of the term "geometry".
> We take the reviewer's point that some uses of the term may feel heavy-handed where the underlying operations are standard linear algebra. That said, we use the term deliberately to emphasize that the **direction and magnitude** of vectors in the attention computation matter for understanding token contribution, not just the scalar attention weights. We will review the manuscript and ensure the term is used where it genuinely aids understanding rather than as a blanket descriptor.
>
> ### Q1: Contribution weights as a new mechanism and future directions.
> We would like to clarify to the reviewers that **contribution weights are not a new model or mechanism** but are simply a new way of looking at what attention is already doing. The attention mechanism aggregates information through attention weights, value vectors, and the output projection. Existing methods look at these factors in isolation. Contribution weights simply **look at all three together**, capturing how attention weight magnitude, value norm, and value-output alignment jointly determine a token's actual influence on the output. Looking forward though we, like the reviewer, are excited about using the insights contribution weights provide to inform architecture design. One concrete direction is investigating whether models can be designed to **avoid producing sink tokens entirely** while still preserving the suppression of uninformative, low-norm outputs that sinks currently provide.

---

> > ### Author Rebuttal · Reviewer_KUpN · 2026-04-06
> >
> > My main concern was the need for some conceptual clarification. The authors have clarified these concepts, so my concerns have been resolved. I have increased my score.

---

### Official Review · Reviewer_VneS · 2026-03-15

**Soundness:** 3
**Presentation:** 3
**Significance:** 2
**Originality:** 2
**Overall Recommendation:** 4
**Confidence:** 4

**Summary:**

This paper introduced **contribution weights** metric which captures the token-wise importance more faithfully than attention-weights. Contribution weights analyze the token importance by taking three factors into account--attention weights, relative norm, and cosine similarity; and provide a geometric notion of token-importance.

Authors have argued that their contribution weight metrics   provide more faithful measure of semantically-critical tokens across different models/architectures, datasets, and downstream task.   They further provide a meaningful interpretation of  attention sink tokens, and their role into suppressing the semantic drift of low-confidence tokens.

**Compliance With Llm Reviewing Policy:**

Affirmed.

**Ethical Review Concerns:**

The ``year'' in most of the references are omitted. Whether it is intentional, to save the space in the paper and cram more text into the main paper, or the issues with bib-style. Authors should clarify whether they have modified the bib-style file of ICML template.

**Ethical Review Flag:**

Flag this paper for an ethics review.

**Ethics Expertise Needed:**

["Other Expertise"]

**Final Justification:**

I have raised the score to 4, based on the rebuttal discussion, particularly how authors' metrics and value-geometry mitigate the bind spots which conventional entropy-based metric cannot explain.


Nonetheless, I believe that including some explanation to attention-induced Rank Collapse through value geometry could make paper much stronger, but the central focus of current manuscript seems confined to active role of attention sink.

**Key Questions For Authors:**

see the weakness

**Limitations:**

Yes

**Strengths And Weaknesses:**

# Strengths

1. The compositional of authors proposed metric provides a broader and more intuitive explanation to various attention-related phenomenon, which attention-weight based metric alone cannot explain otherwise.

2. The contribution weight metric explain the benign role of attention sink token into suppressing semantic drift, which seems unknown.

3. Experiments are conducted on >7B scale models across distinct architectural family: Mistral-7B, LLama *B, Qwen8B and DeepSeek-7B models




# Weaknesses

1. The explanation unlocked by the contribution weight metric seems **just another explanation for attention sink**. There is a solid line of work which already explain the attention sink token, including their geometric interpenetration [1] .

2. Attention-weights based metric such as attention-entropy has been shown to offer meaningful explanation of a range of learning dynamics [2]. It does become obvious how the contribution weight can be incorporated to any unexplained attention phenomenon, which is not yet explained by any existing  metric.

3. Also, how the contrition weights can be operationalize? That is, how this metric can be used as causal signal for better model design, unlike the attention-entropy metric which can be used as active signal [2,3].

4. What is the operational  (including the memory and compute) complexity of computing the contribution metric?


[1] Ruscio et al., What are you sinking? A geometric approach on attention sink, NeurIPS 2025

[2] Jha et al., AERO: Entropy-Guided Framework for Private LLM Inference

[3] Petar Veliˇckovi´c et al., Softmax is not Enough (for Sharp Size Generalisation), ICML 2025

---

> ### Author Rebuttal · Authors · 2026-03-30
>
> We thank the reviewer for appreciating that contribution weights capture phenomena "which attention-weight based metric alone cannot explain otherwise." We believe the concerns raised conflate our value-geometric findings with existing attention-only analyses, and clarify each point below.
> ### W1: "Just another attention sink explanation."
> We respectfully disagree. Existing work, characterizes sinks as passive (Barbero 2025, Guo 2024): high attention, low value norms, effectively performing a null operation. **Our contribution is qualitatively different.** We show that sink value vectors are **actively anti-aligned with semantic tokens** (Figure 5a, Section 6.2) and that this opposition serves a functional role, suppressing semantic drift via a **convex geometric relationship** between sink rate and output norm (Equation 20, Figure 5b), not simple monotonic decay. The causal intervention in Figure 5c confirms this directly. **Reducing sink value norms increases perplexity**, demonstrating active influence rather than passive absorption. Figure 6 shows that without sinks, low-norm head outputs collapse into the semantic cone, introducing noisy drift along the semantic axis. These findings, active directional opposition, convex suppression geometry, and drift cancellation, are **new and invisible to prior metrics** including the geometric framing in (Ruscio 2025).
>
> ### W2: Attention entropy already explains phenomena.
> Firstly, attention entropy and contribution weights measure fundamentally different things. Attention entropy is a summary statistic that measures how spread out or concentrated the attention distribution is, but it operates solely on the attention matrix. **Attention entropy tells you nothing about how much individual tokens actually contribute to the output.**
>
> Secondly, contribution weights capture geometric properties of value vectors that no attention-based metric can access, and this is precisely what allows us to uncover the novel findings in Section 6. We show that sink tokens are **actively anti-aligned** with semantic tokens (Figure 5a) and that this directional opposition serves a functional role, suppressing semantic drift through a **convex geometric relationship** (Equation 20, Figure 5b). The causal intervention in Figure 5c confirms this by showing that reducing sink value norms increases perplexity, **directly falsifying the passive absorber hypothesis**. These findings **depend entirely on value vector geometry** and are invisible to attention entropy or any attention-only metric.
>
> In fact, computing entropy over contribution weights rather than attention weights reveals an interesting phenomenon in its own right. As we show in Appendix D.5 (Table 5), attention entropy for LLaMA-8B (excluding sinks) is S=2.61, while contribution weight entropy is notably lower at S=2.27. This tells us that **value-output alignment acts as a geometric filter** that concentrates effective token influence: tokens with high attention but poor alignment contribute less than their attention weight suggests, while well-aligned tokens contribute more. This sharpening effect is completely undetectable when computing entropy over the attention distribution alone. Table 1 and Appendix B (Figure 7) corroborate this, confirming empirically that **high attention does not imply high contribution**.
>
> ### W3: Operationalization.
> This work focuses on a) introducing the measure with empirical evidence of its faithfulness (Section 5, Figures 3-4), and b) using it to analyze phenomena undetectable by existing attention based metrics (Section 6). We believe both are **substantive contributions in their own right**. However, we agree that using contribution weights for architecture design is an exciting direction for future work. One concrete avenue is designing models that avoid producing sink tokens while still preserving the geometric suppression of low-norm outputs. Contribution weights give us the diagnostic tools to formulate such questions for the first time.
>
> ### W4: Computational complexity.
> Contribution weights require **a single forward pass** and no backward passes like interpolation paths, or iterative computation. The overhead consists of two steps: 1) materializing projected value vectors $\tilde{\mathbf{v}}^{(h)}_j = \mathbf{v}^{(h)}_j W^{(h)}_O$, costing $O(Ld^2)$ per layer (equivalent to the MHA output projection), and 2) computing inner products $\langle \mathbf{o}_i, \tilde{\mathbf{v}}^{(h)}_j \rangle$ for all causal pairs and heads, costing $O(HL^2d)$ per layer. The dominant cost is step 2), a factor of $H$ over the $QK^T$ computation ($O(L^2d)$) since output vectors are $d$-dimensional rather than $d_h$-dimensional. Additional memory is $O(HLd)$ for projected values and $O(HL^2)$ for the contribution matrix per layer. This is modest given the computation is **fully parallelizable with no gradients**, unlike gradient-based attribution which requires one or more backward passes.

---

> > ### Author Rebuttal · Reviewer_VneS · 2026-04-03
> >
> > Thank you for highlighting the blind spots which attention-entropy cannot detect, and also acknowledging the  limitations.
> >
> > I have one follow up question: how would value geometry (and/or contribution weights)  explain the attention-induced **Rank Collapse**--a well known phenomenon [1,2]? Does authors' proposed metrics and value-geometry analysis provide new insights into why self-attention only network collapse to token-uniformity with depth ?
> >
> >
> >
> > [1] Dong et al., Attention is not all you need:pure attention loses rank doubly exponentially with depth, ICML 2021
> >
> > [2] Saada et al., Mind the Gap: a Spectral Analysis of Rank Collapse and Signal Propagation in Attention Layers, ICML 2025

---

> > > ### Author Response · Authors · 2026-04-04
> > >
> > > We thank the reviewer for this thought-provoking question, which points to a natural connection between our work and the rank collapse literature. We give some initial thoughts on the topic below.
> > >
> > > We believe the value geometry perspective offered by contribution weights provides a complementary lens on rank collapse. Dong et al. (2021) show that in pure self-attention networks without skip connections or MLPs, all token representations converge to a single shared direction. Contribution weights allow us to observe this process directly through the alignment factor. As tokens converge, their value vectors align and cosine similarities approach 1 across all pairs. This suggests that the cosine alignment terms in contribution weights could serve as a per-head diagnostic for detecting the early stages of rank collapse, before full convergence occurs.
> > >
> > > We stress though that we have not empirically investigated this connection but consider it a promising direction for future work, particularly in understanding how skip connections and MLPs affect value vector alignment across depth. In fact, analysing depth-dependent properties of transformers using contribution weights is follow up work which we have already begun to look at and we are keen to include such analysis. Once again we thank the reviewer for highlighting this link.

---

### Official Review · Reviewer_dT4K · 2026-03-18

**Soundness:** 3
**Presentation:** 3
**Significance:** 3
**Originality:** 3
**Overall Recommendation:** 5
**Confidence:** 4

**Summary:**

The authors propose Contribution Weights to more accurately estimate the contributions of context tokens, as attention weights alone are not enough to measure token importance in transformers. The method introduced the idea of "value vector", where a token can receive high attention but still contribute little if its value vector is orthogonal to the output. The authors showed that this method gives more faithful token-importance estimation compared to other attention-based or norm-based methods. The author also provide interpretation of attention sinks, where sink tokens receive a large amount of attention weights but are actually anti-aligned with the output direction to suppress semantic drift.

**Compliance With Llm Reviewing Policy:**

Affirmed.

**Final Justification:**

The final rebuttal didn't address my concerns about additional experiments or larger-scale models. My idea for this paper is more neutral now. But I think it is still above the acceptance threshold.

**Key Questions For Authors:**

Do you have the experiments for larger size models, or non-LLaMA family models on the downstream tasks?

**Limitations:**

I see the author mentioned the limitation of Existing Contribution Measures, but did not discuss their proposed method.

**Strengths And Weaknesses:**

Strengths:

- The paper explains why attention-only analysis can be misleading when vectors interfere constructively or destructively, and why alignment is necessary for measuring actual contribution.
- The experiment results showed that the proposed Contribution Weights outperform attention-based baselines in the token-removal experiments across different models and datasets, and it also preserves better downstream accuracy under token removal settings.

Weaknesses:
- Most experiments results are on four 7B–8B decoder-only models, and the downstream-task evaluation is only done with LLaMA-3.1-8B, so it's not clear whether the conclusions can generalize to larger models
- The experiments are dependent on the assumption that "if removing a token is more destructive, the more important the token is". I wonder whether it is always reasonable. For example, a token T appears many (=N) times in different places in the context, so removing T less than N-1 times doesn't really prevent the model from copying and pasting the same T to output. In such a case, other non-important token, let's say T', which only exists once in the context, can have more destructive effects when T' is removed. In short, the interaction between tokens is also important.

---

> ### Author Rebuttal · Authors · 2026-03-30
>
> We greatly appreciate the reviewer spending time reviewing our work and for understanding that "alignment is necessary for measuring actual contributions" one of the core messages of our work. We hope to answer the reviewers remaining queries below.
>
> ### W1: Only Decoder-only models.
> For this work we focused on decoder only models as they are the dominant LLM architecture. We stress however, that our definition of contribution weights is general and is in fact applicable to any model that performs sequence mixing as a weighted sum of tokens, such as encoder-decoder models, bidirectional attention, cross attention, linear attention and state space models. We thank the reviewer for raising this point of clarification and we will make it explicit in the revision the general applicability of contribution weights to other architectures, leaving it to future work to use contribution weights to explore and understand other architectures
>
> ### W2: Larger / non-LLaMA models on downstream tasks.
> We strive to use a wide range of models in our analysis of contirbution weights. In particular our token importance experiments (Section 5) span four model families: LLaMA, Mistral, Qwen, and DeepSeek (Figures 3, 8-10). For our later mechanistic analysis in Section 6 we use LLaMA as it is among the most widely adopted open-weight model families with well-understood training procedures, making it a natural testbed for reproducible interpretability research. Regarding scale, 8B models were the largest we could inference without special techniques on our available compute (3090s and 40GB A100s). Contribution weights add minimal overhead to a single forward pass, so there's no inherent barrier to scaling, we simply prioritized thorough evaluation at accessible sizes over coverage across many model scales.
>
> ### W3: Ablation validity and token interactions.
> We thank the reviewer for their interesting take here. If a token appears N times, any single copy genuinely is less important, and a faithful metric should reflect that. Contribution weights naturally distribute credit across positions carrying similar information, since they are derived from context-dependent attention patterns that already account for redundancy. The reviewers concern about non-additive token interactions is a property of the ablation evaluation protocol, not of our method. As such all baselines face it equally, and relative comparisons remain valid.

---

> > ### Author Rebuttal · Reviewer_dT4K · 2026-04-03
> >
> > Thanks for the reply. Actually I didn't mean to criticize "decoding-only" settings, I also don't think it's necessary to try encoder-decoder model which is no longer used by the community.
> >
> > For non-LLaMA models comments, I mainly asked for "downstream tasks" experiment to be done outside of the llama family models, not just the token importance experiments.
> >
> > For the model scale comments, it should be easy to inference 70B models with device map and cpu offloading with A100s. (https://huggingface.co/docs/accelerate/v0.22.0/en/concept_guides/big_model_inference) It's not even a problem to load Llama-70B into 8*V100s.

---

> > > ### Author Response · Authors · 2026-04-04
> > >
> > > We thank the reviewer for their continued engagement and helpful suggestions and for clarifying their previous stance on decoder-only models which we misinterpreted.
> > >
> > > Regarding downstream task experiments on non-LLaMA models, we appreciate the suggestion. We note that our token importance experiments (Figure 3) already cover Mistral, Deepseek, Qwen and LLaMA, and the faithfulness results are consistent across all three families. We would expect the downstream results to follow the same pattern but agree this would provide additional confirmation.
> > >
> > > Regarding scaling to 70B models, we appreciate the pointer to CPU offloading. We note however that computing contribution weights requires additional memory of $O(HLd)$ for projected value vectors and $O(HL^2)$ for the contribution matrix per layer (Section 4, W4), where d is the full model dimension. For a 70B model with 64 heads, $d=8192$, $l=512$ and 80 layers, this represents a significant overhead beyond standard inference memory. That said, we agree that extending to larger models would strengthen the work and we will investigate memory-efficient strategies that make this feasible.
> > >
> > > We once again thank the reviewer for their constructive feedback and support for our work.

---

### Official Review · Reviewer_FMzK · 2026-03-23

**Soundness:** 3
**Presentation:** 3
**Significance:** 3
**Originality:** 3
**Overall Recommendation:** 4
**Confidence:** 3

**Summary:**

The paper introduces Contribution Weights, a projection-based metric for measuring token importance in transformer self-attention. Contribution weights decompose token influence into three multiplicative factors: attention weight, relative value norm, and cosine alignment with the output direction. The paper claims this metric more faithfully measures token importance than attention-based metrics, and uses it to argue that sink tokens are active information suppressors rather than passive attention absorbers.

**Compliance With Llm Reviewing Policy:**

Affirmed.

**Final Justification:**

I agree that the mechanistic analysis of sink tokens provides meaningful additional contribution beyond the metric itself. While I still view the core decomposition as somewhat incremental, the analytical insights strengthen the overall novelty. Although a comparison to gradient-based methods would further support the faithfulness claim, I accept the authors’ argument that the methods capture different notions.

**Key Questions For Authors:**

1. Does the predicted $\alpha_{\min}$ from Eq. 20 quantitatively match empirically observed sink rates across layers and heads? A scatter plot of predicted vs. observed values is necessary to validate the mechanistic model.

2. Why are gradient based attribution methods excluded from the baseline comparison? These are standard in the interpretability literature and directly address causal token influence.

3. The bottom row of Figure 3 shows larger $\Delta$PPL than the top row at ~80% removal. Does the paper have an explanation for this reversal, and does it affect the validity of the faithfulness evaluation?

4. What is the exact threshold used to identify sink tokens, and how sensitive are the results to this choice?

5. The regression in Table 1 fits a linear model through the origin. Was this fit per layer, per head, or pooled? Pooling across layers and heads could conflate structural variation with within head variation and inflate or deflate $R^2$ values.

**Limitations:**

The paper doesn't include a discussion on the limitations and potential negative societal impact. A dedicated section for this would be most welcome.

**Strengths And Weaknesses:**

**Strengths**

1. The factorization $c_{ij} = \alpha_{ij} \cdot \frac{\|\bar{v}_j\|}{\|\mathbf{o}_i\|} \cdot \cos(\mathbf{o}_i, \bar{v}_j)$ follows naturally from the geometry of weighted vector summation. Figure 2 concretely shows how the three components distribute differently from raw attention weights.

2. Scaling sink value vectors by $\gamma \in [0,1]$ and observing monotonically increasing $\Delta$PPL (Figure 5c) is a direct causal test that falsifies the passive absorber hypothesis.

**Weaknesses**

1. The core definition, projecting a weighted value vector onto the output direction, is a standard inner product operation. The decomposition into attention, norm, and cosine follows trivially from the cosine identity. The paper does not clearly articulate what is fundamentally new beyond applying this decomposition to attention analysis. The incremental gap over VW ON (which already accounts for value norm and output norm) reduces to adding a single cosine term, and the paper does not make a compelling case that this warrants a standalone contribution.

2. The quadratic model in Eq. 20 collapses all non sink tokens into a single aggregate vector with uniform mutual alignment, a strong simplifying assumption. The paper derives $\alpha_{\min} < 1$ as the key mechanistic claim but never verifies whether the predicted $\alpha_{\min}$ matches empirically observed sink rates across layers and heads. Without this, the central mechanistic argument remains qualitative and the model is unfalsified.

3. At high removal rates (~80%), the bottom row of Figure 3 (removing unimportant tokens) shows larger $\Delta$PPL than the top row (removing important tokens) at the same removal fraction. Removing important tokens should hurt more than removing unimportant ones. The paper never acknowledges or explains this reversal. This is a direct inconsistency in the faithfulness evaluation that undermines the central empirical claim.

4. Sinks are identified via near perfect collinearity with [BOS] but the exact threshold is never specified. Since sink tokens are excluded from the token removal experiments, this choice directly affects the faithfulness evaluation and the results cannot be reproduced without a precise specification.

5. The paper positions contribution weights against raw attention, VW, and VW ON, but does not compare against gradient based or integrated gradient attribution methods, which directly address the causal influence question. The absence of these baselines overstates the relative performance of contribution weights.

6. All experiments use autoregressive decoder only models. The paper does not discuss whether the findings generalize to encoder or encoder decoder architectures, limiting the claimed generality of the contribution.

---

> ### Author Rebuttal · Authors · 2026-03-30
>
> We thank the reviewer for their detailed engagement with our work. We address each point below. We that believe several concerns stem from misunderstandings which we hope to clarify.
>
> ### W1: Core definition is standard / incremental over VW-ON.
> In Section 4.1 we show that three factors independently determine each token's contribution to a weighted vector sum: attention weight, value norm, and relative alignment. **No prior method accounts for all three.** Appendix B demonstrates this concretely using LLaMA-8B showing that: a) weight-based analysis ignores variation in value magnitudes, b) norm-based analysis ignores value-output alignment, and c) distance-based analysis cannot distinguish constructive from destructive interference (Figure 7).
>
> Further, we reject the claim that the cosine term is merely incremental over VW-ON. First, contribution weights outperform all baselines for all models, on both comprehensiveness and sufficiency, and on both language modelling (Figure 3) and downstream tasks (Figure 4). Second, **VW-ON is itself a contribution of this paper**, introduced by us to capture head-level importance. The reviewer appears to have confused it with existing work. Third, the regression in Table 1 shows that for sink tokens, cosine similarity alone explains $R^2 = 0.518$ of contribution variance while attention explains only $R^2 = 0.079$. For non-sink tokens, attention combined with cosine ($R^2 = 0.834$) substantially outperforms attention combined with norm ($R^2 = 0.579$), which is what existing approaches rely on.
>
> ### W2: Quadratic model assumptions and $\alpha_{\min}$ not verified.
> The quadratic model in Eq. 20 relies on two empirical assumptions (Appendix E): that $\|w\| \approx \kappa(1 - \alpha)$ and $\langle v_0, w \rangle \approx \gamma(1 - \alpha)$. Figure 11 (top) validates these with consistently high $R^2$ scores (median > 0.8 across most layers). The model's purpose is to explain the **convex shape** of the sink-rate vs output-norm curve (Figure 5b), and $\alpha_{\min} < 1$ is a direct mathematical consequence of this validated convexity.
>
> The reviewer is right that a direct comparison of predicted vs true $\alpha_{\min}$ strengthens the argument. We have generated a scatter plot which shows a clear positive relationship, with the theoretical minimum systematically **overestimating** the true minimum. This actually strengthens our central claim: the true minimum is even further below 1 than predicted, firmly falsifying the view that minimum output norm occurs at $\alpha = 1$. We will add this plot to the appendix.
>
> ### W3: Figure 3 bottom row shows larger $\Delta$PPL than top row.
> The reviewer is comparing across **different x-axis scales**. The top row (comprehensiveness) removes 0.25% to 1% of the most important tokens. The bottom row (sufficiency) removes 40% to 90% of the least important tokens. There is no inconsistency. Using contribution weights, removing just $\sim$0.5% of important tokens causes a comparable $\Delta$PPL ($\sim$1) to removing $\sim$80% of unimportant tokens. This is exactly what a faithful metric should show, that a small fraction of tokens carry most of the functional weight.
>
> ### W4: Sink identification threshold.
> Sink tokens are identified by cosine similarity > 0.95 with the [BOS] token representation. We will specify this threshold explicitly in the revision.
>
> ### W5: No gradient-based baselines.
> Gradient-based methods measure **sensitivity** (how the output changes under perturbation), not how information is routed through the forward pass. The baselines we include (attention, VW, VW-ON) are the correct comparison class: single-forward-pass, architecture-derived decompositions. Integrated gradients require multiple forward passes along an arbitrary interpolation path and have well-documented pathologies including gradient saturation (Adebayo et al., 2018; Hooker et al., 2019). Contribution weights also provide a lens into value vector alignment (Section 6), which gradient methods cannot support. We will add a discussion clarifying this distinction.
>
> ### W6: Only decoder-only models.
> We focused on decoder-only models as they are dominant LLM architecture. However, we note that contribution weights depend only on the structure of the attention mechanism, and hence are generalizable to any attention based architecture such as encoder and encoder-decoder architectures. We will clarify this and note encoder/encoder-decoder validation as future work. Please also see also our rebuttal to W1 in Reviewer dT4K for further details.
>
> ### Q5: Table 1 regression pooling.
> As described in Appendix D.2, we sample 5 million data points across all layers, heads, and positions, with 5-fold cross-validation on a held-out set. Regressions are pooled but **separated by token group** ($j = 1$, $j > 1$, all tokens) to avoid conflating structurally different populations. Results are consistent across FineWeb-Edu and GSM8K (Table 3), suggesting robustness to dataset choice.

---

> > ### Author Rebuttal · Reviewer_FMzK · 2026-04-04
> >
> > Thank you for the detailed rebuttal. I have read it carefully. While several concerns have been adequately addressed, a few key issues remain.
> >
> > - W1 (Partially Resolved). I acknowledge the clarification that VW-ON is introduced as the authors’ contribution. However, the concern regarding incremental novelty remains only partially addressed. The core operation, i.e. projecting weighted value vectors onto the output direction, effectively reduces to a standard inner product. As such, the framing as a distinct geometric insight appears somewhat overstated relative to its incremental difference from existing norm-based approaches.
> >
> > - W5 (Unresolved). While the rebuttal argues that gradient-based methods quantify a different notion, the absence of empirical comparison makes it difficult to assess whether the proposed contribution weights provide more faithful or practically superior explanations of causal token influence. Given that faithfulness is a central claim of the paper, this omission remains significant.
> >
> > - W6 (Partially Resolved). Deferring validation on encoder and encoder-decoder architectures to future work is reasonable. However, this limitation reduces the strength of the generality claims currently made in the abstract and conclusion.
> >
> > Overall, since W1 and W5 remain substantively unresolved, I'll update my score to 3.

---

> > > ### Author Response · Authors · 2026-04-04
> > >
> > > We thank the reviewer for their thoughtful reassessment and for raising their score. We believe a few points merit further discussion, as we feel the scope of our contributions may not be fully reflected in the current evaluation.
> > >
> > > **W1.** We respectfully believe the reviewer may be underweighting the contribution due to the simplicity of contribution weight decomposition. We would note that many impactful analytical tools in machine learning are grounded in straightforward mathematical operations. The value lies in identifying the right decomposition and demonstrating its utility. To probe this further, **would the reviewer consider existing norm-based approaches, which simply multiply attention scores by the norm of the value vector, to also constitute only "incremental novelty"?** If not, we would argue that incorporating alignment, which Table 1 shows explains substantially more variance than norm alone, warrants at least comparable recognition.
> > >
> > > More importantly, we emphasise that the contribution of this paper extends well beyond proposing a metric. The faithfulness evaluation (Section 5) is only one half of the paper. Section 6 uses contribution weights to analyse the functional role of attention and value vector geometry with respect to sink tokens, an analysis that cannot be conducted using attention weights, norm-based methods, or gradient-based methods. Reviewer bUeb, for example, described our sink token analysis as providing _"an important piece of the puzzle for understanding attention sinks."_ **We would welcome the reviewer's assessment of how the analytical contributions of Section 6, towards understanding attention sinks, factor into their evaluation of novelty?**
> > >
> > > **W5.** We appreciate the reviewer's continued emphasis on gradient-based baselines. If the sole contribution of this work were a more faithful token importance metric, we agree that such a comparison would be warranted. However, as noted above, the experimental work is divided into two parts: (1) demonstrating that contribution weights are in many cases considerably more faithful than attention and norm-based approaches, and (2) using contribution weights as an analytical lens to study the geometry of value vectors and the mechanistic role of sink tokens. We would stress that gradient-based methods and contribution weights answer fundamentally different questions. Gradient methods ask "what happens if I perturb this token?", while contribution weights ask "what are the geometric factors that determine this token's influence on the output?" As a result, **gradient-based methods cannot provide the compositional, per-head decomposition into alignment, norm, and attention that contribution weights can and which drives the analysis in Section 6.** That said, for the faithfulness evaluation in Section 5 specifically, we are open to including a comparison with gradient-based methods in the camera ready if the reviewer considers it essential for acceptance.
> > >
> > > **W6.** We appreciate the reviewer's point, but note that decoder-only models are the dominant architecture in contemporary language modelling. Encoder-only models such as BERT (Devlin et al., 2019) were introduced nearly eight years ago, and the field has largely moved toward autoregressive architectures. Reviewer dT4K similarly noted that they _"don't think it's necessary to try encoder-decoder model which is no longer used by the community."_ We will temper the generality claims in the abstract and conclusion to reflect the decoder-only scope, while noting that contribution weights depend only on the structure of the attention mechanism and are in principle applicable to any attention-based architecture.

---

### Decision · Program_Chairs · 2026-04-30

**Decision:**

Accept (regular)

**Comment:**

This paper introduces Contribution Weights, a projection-based metric that decomposes token influence into attention weight, value norm, and cosine alignment with the output direction, and uses it to reveal that attention sink tokens actively suppress semantic drift rather than passively absorbing excess attention. Reviews are positive (5/5/4/4/4, after Reviewer FMzK raised from 3 to 4 during discussion). Three reviewers marked concerns as fully resolved, and Reviewer bUeb described the sink token analysis as "an important piece of the puzzle." The main criticism, raised by Reviewer FMzK, is that the core decomposition is mathematically straightforward and incremental over norm-based approaches. This is fair as a description of the metric alone, but the paper's value lies in applying the decomposition to produce novel mechanistic findings about sink tokens, including directional opposition, convex suppression geometry, and causal validation via value norm interventions. These findings are inaccessible to attention-only or gradient-based methods. The absence of gradient-based baselines and the restriction to decoder-only models at 7B to 8B scale are noted limitations but do not undermine the core contributions. The authors should temper generality claims, specify the sink identification threshold explicitly, and consider adding downstream evaluation on non-LLaMA models